# QUANTIZATION WITH PURPOSE: LOSS-AWARE BIT ALLOCATION FOR GRADIENT COMPRESSION

## ABSTRACT

Gradient quantization is a critical technique for reducing communication overhead in large-scale distributed training. However, existing methods often employ fixed bit-width quantization or adaptive quantizers optimized with signal-level distortion metrics such as MSE, which poorly correlate with model performance. In this paper, we propose a novel layer-wise bit allocation framework for gradient quantization, formulated under a rate-distortion optimization (RDO) paradigm. Unlike prior approaches, our method introduces a loss-aware distortion metric that directly quantifies the impact of quantization on training loss, enabling task-aligned solution for bit allocation. A key insight of our work is the linear superposition property of cross-layer loss distortion, which we theoretically justify and empirically validate. This property allows us to decouple the original joint optimization problem and efficiently solve it via a Lagrangian optimization algorithm with linear complexity. Extensive experiments across vision and language tasks—using CNNs, ViTs, LSTMs, and Transformers—demonstrate the effectiveness of our approach. Moreover, our method integrates seamlessly with existing gradient compression techniques, yielding consistent performance gains.

## 1 INTRODUCTION

The rapid development of large-scale deep learning models has significantly reshaped the landscape of artificial intelligence. Foundation models such as GPT-4 (Achiam et al., 2023) and large-scale diffusion models like Stable Diffusion (Rombach et al., 2022) have demonstrated impressive generalization capabilities across a broad range of tasks. However, the enormous number of parameters in these models, often reaching hundreds of billions, necessitates distributed training across large GPU clusters. Within this paradigm, the synchronization of gradients or model parameters incurs substantial communication overhead, which has emerged as a primary bottleneck that limits both scalability and training efficiency (Wang et al., 2023; Tang et al., 2021).

To alleviate this bottleneck, gradient compression has become an essential component of modern distributed training systems. Existing approaches can be broadly categorized into three classes: sparsification (Stich et al., 2018; Lin et al., 2017; Wangni et al., 2018), low-rank decomposition (Vogels et al., 2019; Wang et al., 2018; Yu et al., 2018), and quantization. Among these, quantization is particularly appealing due to its simplicity and compatibility with contemporary hardware, making it a central focus of this study. Early studies, such as TernGrad (Wen et al., 2017) and QSGD (Alistarh et al., 2017), employed uniform quantization across fixed ranges, while more recent methods like Natural Compression (NC) (Horvóth et al., 2022) improved fidelity through non-uniform schemes. However, a common limitation of these methods is that their quantization levels are statically defined at the beginning of training and remain fixed throughout the training process, which is often suboptimal in practice due to the dynamic nature of gradient distributions. To address this, adaptive quantization methods have been investigated to dynamically adjust quantization strategies in response to the evolving characteristics of the training process. Prior work has primarily focused on intra-layer adaptation, where quantization parameters, such as clipping range or step size, are adjusted over time. However, these approaches typically overlook inter-layer heterogeneity, treating all layers uniformly despite their differing sensitivities to quantization. Representative methods, including AdaQS (Guo et al., 2020), AQG (Mao et al., 2022), and ALQ (Faghri et al., 2020), introduce various mechanisms for adaptive adjustment of quantization parameters. While these techniques

offer increased flexibility, they typically employ a uniform bit-width across all layers, thereby over-looking the varying sensitivity of gradients in different layers to quantization.

In light of this, mixed-precision quantization approaches have been explored to assign different bit-widths to gradients based on their relative sensitivity to compression. AC-SGD (Yan et al., 2022) adaptively adjusts the quantization bit-width over training iterations with respect to the norm of gradients, communication budget, and the remaining training steps. L-GreCo (Markov et al., 2024) employs dynamic programming to determine the minimum bit-width for each layer under a global quantization error constraint. Different from these two approaches, our work focuses on solving the layer-wise bit allocation problem under a fixed per-iteration communication budget constraint. Moreover, the above two methods rely on signal-level metrics, such as the L2 norm or mean squared error (MSE) of gradients, to estimate sensitivity to compression, which exhibit weak correlation with the actual impact of quantization on model performance.

In this paper, we propose a layer-wise bit allocation framework for adaptive gradient quantization. Specifically, we formulate the bit allocation problem within a rate-distortion optimization (RDO) paradigm, aiming to minimize task-related distortion under a fixed communication budget. Instead of optimizing signal-level quantization errors (typically heuristic metrics such as L2-norm or MSE), we introduce a loss-aware distortion metric that directly quantifies the impact of quantization on the training objective. This enables fine-grained, dynamic bit allocation guided by true optimization sensitivity. To decouple the RDO problem, we investigate and validate a linear superposition property of cross-layer loss distortion through theoretical and empirical analysis. Specifically, we show that the total degradation in training loss resulting from the joint quantization of multiple layers can be well approximated by the sum of individual distortions incurred when each layer is quantized independently. To solve the decoupled RDO problem, we further develop a Lagrangian optimization algorithm that identifies the optimal bit allocation while reducing computational complexity from exponential to linear in the number of layers. In addition, to efficiently perform bit allocation during model training without introduce unnecessary computational overhead, we propose a dynamic real-location trigger that monitors changes in the gradient distribution and initiates bit assignment when significant shifts are detected.

The primary contributions of this paper are summarized as follows:

- We formulate the bit allocation problem for gradient quantization within a rate-distortion optimization (RDO) framework, which directly optimizes the training loss rather than widely-used signal-level quantization errors, thereby ensuring improved model training performance.

- We theoretically prove and empirically validate a linear superposition property of loss distortion, which enables efficient decomposition of the RDO problem. Based on this property, we propose a Lagrangian optimization method that achieves optimal bit allocation with linear computational complexity.

- We evaluate our method across a diverse set of model architectures, including CNNs, Vision Transformers (ViTs), Transformers, and LSTMs, on both image classification and language modeling tasks. Experimental results demonstrate that our approach consistently outperforms state-of-the-art static and adaptive quantization baselines.

## 2 METHODOLOGY

### 2.1 PROBLEM FORMULATION: AN RDO PERSPECTIVE

Gradient quantization aims to transform a full-precision gradient $g$ into a lower-bit representation $\tilde{g}$ to reduce communication overhead during distributed training. In layer-wise bit allocation problem, the key challenge is to determine how to distribute a limited communication budget $B_{\text{total}}$ (measured in bits) across the layers of a neural network. Consider a model with $L$ layers, where $\mathbf{g}^{(l)}$ denotes the gradient tensor of layer $l$, and $N_l$ is its number of parameters. The objective is to assign a bit-width $b_l \in \mathcal{B}_{\text{options}}$ to each layer such that the total quantization-induced distortion is minimized, subject to the overall budget constraint.

We formulate this as a classic RDO problem, where the "rate" $R$ refers to the total number of bits used for communication, and the "distortion" $D$ quantifies the performance degradation due to

quantization. The optimization objective is given by:

$$\min_{\{b_l\} \in \mathcal{B}_{\text{options}}} D_{\text{total}} = D(\mathbf{g}, Q(\mathbf{g}, \{b_l\})), \quad \text{s.t.} \quad R_{\text{total}} = \sum_{l=1}^{L} b_l \cdot N_l \leq B_{\text{total}}, \tag{1}$$

where $Q(\mathbf{g}, \{b_l\})$ denotes the quantization function applying $b_l$ bits to $\mathbf{g}^{(l)}$. To secure final training performance, the optimization objective (i.e., the distortion metric) should be carefully designed.

### 2.1.1 THE LOSS-AWARE DISTORTION METRIC

Traditional distortion metrics, such as MSE, quantify the geometric deviation between the original and quantized gradients. While computationally efficient, these metrics often exhibit weak alignment with the ultimate objective of deep learning optimization: minimizing the training loss. Therefore, to better capture the true impact of quantization on model performance, we propose a Loss-Aware Distortion (LAD) metric, which directly measures the change in the training objective induced by quantizing the gradient of a specific layer.

Let $\mathbf{W}_t$ denote the model weights at iteration $t$ and $\eta$ denote the learning rate. Consider a set of $K$ data batches $\mathcal{S} = \{d_1, d_2, \ldots, d_K\}$ sampled from the training data, we first define the per-batch loss difference $\Delta\mathcal{L}_l(b_l; d)$ for layer $l$ with bit-width $b_l$ as:

$$\Delta\mathcal{L}_l(b_l; d) = \left| \mathcal{L}(\mathbf{W}_t - \eta \cdot \mathbf{g}_{\text{mixed},t}^{(l)}; d) - \mathcal{L}(\mathbf{W}_t - \eta \cdot \mathbf{g}_t; d) \right|, \tag{2}$$

where $\mathbf{g}_t = (\mathbf{g}_t^{(1)}, \ldots, \mathbf{g}_t^{(L)})^T$ is the gradient vector of all $L$ layers and $\mathbf{g}_{\text{mixed},t}^{(l)} = (\mathbf{g}_t^{(1)}, \ldots, \tilde{\mathbf{g}}_t^{(l)}, \ldots, \mathbf{g}_t^{(L)})^T$ is a mixed gradient vector where only the gradient for layer $l$ (i.e. $\mathbf{g}_t^{(l)}$) is replaced by its quantized version $\tilde{\mathbf{g}}_t^{(l)}$. We then define the Loss-Aware Distortion $D_l(b_l)$ as the expected loss difference over the batch set $\mathcal{S}$:

$$D_l(b_l) = \mathbb{E}_{d \in \mathcal{S}} \left[ \Delta\mathcal{L}_l(b_l; d) \right]. \tag{3}$$

This expectation over multiple batches mitigates the effects of single-batch noise and yields a more stable and reliable signal for guiding bit allocation.

By quantifying distortion in terms of its impact on the training loss rather than gradient geometry, the proposed metric offers a task-aligned and dynamic measure of gradient sensitivity, better supporting optimization-aware quantization decisions. Our ablation study in Appendix B.1 confirms this, showing that using our LAD metric consistently yields higher accuracy than using a traditional MSE metric.

### 2.2 THE LINEAR SUPERPOSITION PROPERTY

A fundamental challenge in solving the RDO problem (Equation 1) lies in the combinatorial complexity of the joint optimization. Specifically, assigning bit-widths to $L$ layers from a discrete set $\mathcal{B}$ results in a search space of size $|\mathcal{B}|^L$, which grows exponentially with the number of layers. In deep neural networks, where $L$ may reach hundreds or more, this combinatorial explosion renders brute-force approaches (such as heuristic algorithms and greedy search methods) computationally infeasible. Moreover, the total distortion $D_{\text{total}}$ exhibits highly non-linear and coupled dependencies on the joint bit allocation $b_l$, as it reflects complex interactions among quantization errors across layers within the non-convex loss landscape of deep networks. It makes the joint bit allocation problem particularly challenging.

To address this challenge, we introduce and validate a linear superposition property, which asserts that the total change in training loss resulting from the simultaneous quantization of multiple layers can be closely approximated by the sum of the individual distortions incurred when each layer is quantized independently, i.e.:

$$D_{\text{joint}}(\{b_l\}) \approx \sum_{l=1}^{L} D_l(b_l). \tag{4}$$

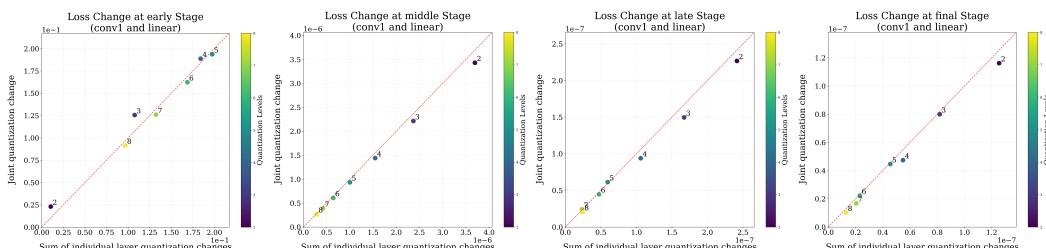

Figure 1: Empirical validation of the Linear Superposition property on ResNet-18. The joint loss change from quantizing two layers simultaneously (y-axis) versus the sum of their individual loss changes (x-axis). Each point represents a different quantization level (2-8 bits), evaluated at various stages of training.

### 2.2.1 THEORETICAL PROOF

We provide a theoretical justification for the linear superposition property based on a first-order Taylor expansion of the loss function. Our analysis is predicated on the following standard assumptions.

**Assumption 1** (Smoothness). *The loss function $L(W)$ is L-smooth, meaning its gradient is L-Lipschitz continuous:*

$$\|\nabla L(W_1) - \nabla L(W_2)\| \leq L\|W_1 - W_2\|, \quad \forall W_1, W_2. \tag{5}$$

**Assumption 2** (Small Perturbation Regime). *The total weight perturbation $\delta W$ induced by quantization is sufficiently small, such that the higher-order terms in the Taylor expansion are negligible compared to the first-order term.*

Assumption 1 implies a bounded Hessian, $\|\nabla^2 L(W)\| \leq L$, and is widely adopted in the analysis of neural networks with smooth activation functions. Since the learning rate $\eta$ is typically small for stable training and any reasonable quantization scheme ensures that the quantization error $\|e^{(l)}\|$ is bounded, often proportionally to the gradient norm $\|g^{(l)}\|$, Assumption 2 is satisfied in practice.

Under these assumptions, we now proceed with the proof. For a given layer $l$, we define the quantization error as $e^{(l)} = (0, \ldots, 0, g^{(l)} - \tilde{g}^{(l)}, 0, \ldots, 0)^T$. When a subset $\mathcal{K}$ of layers is quantized, the resulting perturbation to the weights can be expressed as:

$$\delta\mathbf{W}^{(\mathcal{K})} = (\mathbf{W}_t - \eta\mathbf{g}_{\text{mixed}}^{(\mathcal{K})}) - (\mathbf{W}_t - \eta\mathbf{g}) = \eta\sum_{l\in\mathcal{K}}\mathbf{e}^{(l)}. \tag{6}$$

Let $\mathbf{W}_{\text{orig}} = \mathbf{W}_t - \eta\mathbf{g}$ denote the pristine weight after a full-precision update. According to Equation 2, the distortion induced by quantizing layers in $\mathcal{K}$ is then:

$$D_{(\mathcal{K})} = \mathcal{L}(\mathbf{W}_{\text{orig}} + \delta\mathbf{W}^{(\mathcal{K})}) - \mathcal{L}(\mathbf{W}_{\text{orig}}). \tag{7}$$

Applying a first-order Taylor expansion to $\mathcal{L}(\mathbf{W}_{\text{orig}} + \delta\mathbf{W}^{(\mathcal{K})})$ around $\mathbf{W}_{\text{orig}}$ yields:

$$D_{(\mathcal{K})} \approx \nabla\mathcal{L}(\mathbf{W}_{\text{orig}})^T\delta\mathbf{W}^{(\mathcal{K})} = \mathbf{g}_{t+1}^T\left(\eta\sum_{l\in\mathcal{K}}\mathbf{e}^{(l)}\right), \tag{8}$$

where $\mathbf{g}_{t+1} = \nabla\mathcal{L}(\mathbf{W}_{\text{orig}})$ is the gradient evaluated in updated full-precision weights. Given linearity of the dot product, we can distribute it across the sum:

$$D_{(\mathcal{K})} \approx \sum_{l\in\mathcal{K}}\left(\eta \cdot \mathbf{g}_{t+1}^T\mathbf{e}^{(l)}\right). \tag{9}$$

The term $\eta \cdot \mathbf{g}_{t+1}^T \mathbf{e}^{(l)}$ corresponds to the first-order approximation of the individual distortion $D_l$, i.e., the distortion caused by quantizing only layer $l$. Therefore, we obtain the approximate linear superposition:

$$D_{(\mathcal{K})} \approx \sum_{l \in \mathcal{K}} D_l. \tag{10}$$

The approximation error stems from the higher-order terms in the Taylor expansion. The leading second-order term is given by $(\Delta \mathbf{W})^T \mathbf{H} (\Delta \mathbf{W})$, where $\mathbf{H}$ denotes the Hessian matrix. This term includes interaction effects between different error vectors and is quadratic in both the learning rate $\eta$ and the magnitude of the quantization error, making its contribution negligible in practice.

In addition, the above proof is based on the stochastic gradient descent (SGD). We also demonstrate that the linear superposition property remains valid for the AdamW (Loshchilov & Hutter, 2017) optimizer. The detailed proof is provided in Appendix A.

### 2.2.2 EMPIRICAL VALIDATION

We further provide empirical evidence demonstrating that the higher-order terms in the Taylor expansion are indeed negligible. Figure 1 presents a comparison between the joint distortion $(D_{i,j})$ and the sum of individual distortions $(D_i + D_j)$ for two representative layers, evaluated across bit-widths ranging from 2 to 8. The data points align closely with the $y = x$ diagonal, indicating strong agreement and validating the linear superposition property at various stages of training.

### 2.2.3 PROBLEM DECOUPLING.

Validating this linear superposition property enables the decoupling of the intractable joint optimization problem. By substituting the total distortion with the sum of individual LAD metrics (Equation 3), our objective function becomes separable:

$$\min_{\{b_l\} \in \mathcal{B}_{\text{options}}} \sum_{l=1}^{L} D_l(b_l), \quad \text{s.t.} \sum_{l=1}^{L} b_l \cdot N_l \leq B_{\text{total}}. \tag{11}$$

This transformation reduces the problem's complexity from exponential to linear with respect to the number of layers, enabling practical applicability even in deep neural networks. It serves as the foundation for our proposed bit allocation algorithm.

### 2.3 SOLVE BIT ALLOCATION VIA LAGRANGIAN RELAXATION

To solve the decoupled constrained optimization problem efficiently, we adopt **Lagrangian relaxation** approach. By introducing a Lagrange multiplier $\lambda \geq 0$, we can convert the constrained problem into an unconstrained objective:

$$J(\lambda) = \sum_{l=1}^{L} \left[ D_l(b_l) + \lambda \cdot (b_l \cdot N_l) \right]. \tag{12}$$

The multiplier $\lambda$ can be interpreted as the trade-off factor between distortion $D$ and rate $R$ (i.e., bits). A greater $\lambda$ places a higher penalty on bit usage, favoring lower-bit allocations, while a smaller $\lambda$ prioritizes minimizing distortion. For a fixed $\lambda$, the optimal bit-width for each layer $b_l^*$ can be obtained independently by minimizing its layer-wise Lagrangian cost:

$$b_l^*(\lambda) = \arg\min_{b_l \in \mathcal{B}_{\text{options}}} \left[ D_l(b_l) + \lambda \cdot (b_l \cdot N_l) \right], \tag{13}$$

since the solution space for each layer is small and discrete (e.g., 1 to 8 bits), this step can be performed efficiently.

The remaining challenge is to identify the optimal value of $\lambda^*$ that minimizes total distortion while satisfying the overall communication budget constraint. Given that the total rate $R_{\text{total}}(\lambda)$ is a monotonically non-increasing function of the Lagrange multiplier $\lambda$, we can efficiently identify the optimal value $\lambda^*$ using a bisection search. This process iteratively refines a bounded interval $[\lambda_{\text{low}}, \lambda_{\text{high}}]$, converging to a value that yields the total usage of bits as close as possible to, but not exceeding, the target budget $B_{\text{total}}$. This entire procedure allows us to find an optimal, per-layer bit allocation in a computationally efficient manner. The complete algorithm is described in Algorithm 1.

---

**Algorithm 1** Lagrangian Search for Layer-wise Bit Allocation

---

**Require:** Gradients $\{\mathbf{g}^{(l)}\}_{l=1}^{L}$, bit options $\mathcal{B}$, average quantization bit-width constraint $B_c$, data batches $\mathcal{D}$
**Ensure:** Bit allocation $\{b_l\}_{l=1}^{L}$

1: $B_{total} \leftarrow B_c \cdot \sum_{l=1}^{L} N_l$
2: **Compute R-D curves:**
3: **for** each layer $l \in \{1, \ldots, L\}$ **do**
4:     **for** each bit option $b \in \mathcal{B}$ **do**
5:         $d_{l,b} \leftarrow \text{LossDiff}(g_l, b, \mathcal{D})$ {Distortion per layer}
6:         $r_{l,b} \leftarrow b \cdot N_l$ {Rate per layer}
7:     **end for**
8: **end for**
9: **Estimate $\lambda$ range:** $\lambda_{low}, \lambda_{high}$ from R-D slopes
10: $\mathcal{A}^* \leftarrow \emptyset$
11: **while** $\lambda_{high} - \lambda_{low} > \epsilon$ **do**
12:     $\lambda_{mid} \leftarrow (\lambda_{low} + \lambda_{high})/2$
13:     $B_{used} \leftarrow 0$
14:     **for** each layer $l \in \{1, \ldots, L\}$ **do**
15:         $b_l^* \leftarrow \arg\min_{b \in \mathcal{B}}\{d_{l,b} + \lambda_{mid} \cdot r_{l,b}\}$
16:         $B_{used} \leftarrow B_{used} + b_l^* \cdot N_l$
17:     **end for**
18:     **if** $B_{used} > B_{total}$ **then**
19:         $\lambda_{low} \leftarrow \lambda_{mid}$
20:     **else**
21:         $\lambda_{high} \leftarrow \lambda_{mid}$
22:         $\mathcal{A}^* \leftarrow \{b_l^*\}_{l=1}^{L}$
23:     **end if**
24: **end while**
25: **return** $\mathcal{A}^*$

---

### 2.4 PERFORM BIT ALLOCATION WITH A DYNAMIC REALLOCATION TRIGGER MECHANISM

The previous sections detail how to optimally allocate bit-widths across layers under a fixed communication budget. However, computing the LAD metric (Equation 3) for all layers and candidate bit-widths incurs substantial computational overhead, primarily due to the need for multiple forward passes. Moreover, prior work has shown that the optimal bit allocation tends to remain relatively stable across consecutive training steps (Markov et al., 2024), making full reallocation at every iteration both computationally expensive and largely unnecessary.

This motivates the need for a mechanism to determine when bit reallocation should be performed. To this end, we propose a lightweight yet effective dynamic reallocation trigger that adaptively decides when to recompute the allocation based on changes in gradient statistics. Specifically, we monitor the distribution of L2-norms of the gradients across all $L$ layers, which serves as a compact and computationally efficient proxy for detecting shifts in the overall gradient landscape. Let $\mathbf{v}_t \in \mathbb{R}^L$ be the gradient norm vector at iteration $t$, where the $l$-th element is the L2-norm of the gradient of layer $l$:

$$\mathbf{v}_t = \left[\|\mathbf{g}_t^{(1)}\|_2, \|\mathbf{g}_t^{(2)}\|_2, \ldots, \|\mathbf{g}_t^{(L)}\|_2\right]^T. \tag{14}$$

The trigger mechanism works as follows:

1. **Anchoring:** After a bit allocation is performed at step $t_{\text{alloc}}$, we take the normalized gradient norm vector as an "anchor" vector $\mathbf{v}_{\text{anchor}}$. Normalization ensures that subsequent comparisons reflect changes in the distribution shape rather than overall magnitude:

$$\mathbf{v}_{\text{anchor}} = \frac{\mathbf{v}_{t_{\text{alloc}}}}{\|\mathbf{v}_{t_{\text{alloc}}}\|_2}. \tag{15}$$

2. **Monitoring:** At each subsequent training step $t > t_{\text{alloc}}$, we compute the current normalized gradient norm vector, $\mathbf{v}_{\text{current}} = \mathbf{v}_t / \|\mathbf{v}_t\|_2$.

Table 1: Accuracy recovery and compression ratios for different compression methods with uniform and adaptive schemes on image classification tasks at 2-bit quantization. 'Acc.' refers to Top-1 accuracy and 'Comp. Ratio' refers to compression ratio.

| | ResNet-18 on CIFAR-10 | | ViT-small on ImageNet | | | | | |
| | | | 0–50 epoch | | 150–200 epoch | | 250–300 epoch | |
| Method | Acc.(%) | Comp. Ratio | Acc.(%) | Comp. Ratio | Acc.(%) | Comp. Ratio | Acc.(%) | Comp. Ratio |
|---|---|---|---|---|---|---|---|---|
| FP32 | 88.24 | - | 60.11 | - | 63.24 | - | 65.36 | - |
| Uniform | 77.33 | 16.00× | 47.88 | 16.00× | 62.84 | 16.00× | 65.24 | 16.00× |
| + Greedy | 88.09 | 16.35× | 51.13 | 16.07× | 64.03 | 16.07× | 65.30 | 16.08× |
| + Ours | **88.39** | **16.46×** | **52.66** | **16.28×** | **64.15** | **16.26×** | 65.33 | 16.24× |

Table 2: Accuracy recovery and compression ratios for different compression methods with uniform and adaptive schemes on language modeling tasks at 3-bit quantization.

| Method | LSTM on PTB | | Transformer on WikiText-103 | |
| | Perplexity | Compression Ratio | Perplexity | Compression Ratio |
|---|---|---|---|---|
| FP32 | 82.32 | - | 77.18 | - |
| Uniform | 639.94 | 10.67× | 133.64 | 10.67× |
| + Greedy | 561.79 | 11.27× | 128.20 | 11.51× |
| + Ours | **388.08** | **12.37×** | **118.27** | **12.77×** |

3. **Similarity Check:** We then measure the change between the current and the anchor vector using **Cosine Similarity**:

$$\text{Similarity}(t) = \frac{\mathbf{v}_{\text{current}} \cdot \mathbf{v}_{\text{anchor}}}{\|\mathbf{v}_{\text{current}}\|_2 \|\mathbf{v}_{\text{anchor}}\|_2} \tag{16}$$

The denominator is unity since both vectors are L2-normalized.

4. **Triggering Condition:** A reallocation is triggered at step $t$ when both of the following two conditions are met: (a) the similarity between the current and anchor gradient norm vectors falls below a predefined threshold $\tau$ (e.g., $\tau = 0.92$), indicating a significant shift in the gradient distribution; and (b) a minimum number of iterations, $k_{\min}$, has elapsed since the last allocation at $t_{\text{alloc}}$ (i.e., $t - t_{\text{alloc}} \geq k_{\min}$).

When a trigger condition is met, the full adaptive bit allocation algorithm is executed, and the current gradient norm vector $\mathbf{v}_{\text{current}}$ is stored as the new anchor vector $\mathbf{v}_{\text{anchor}}$ for subsequent iterations. This dynamic trigger mechanism ensures that computational resources for reallocation are used only when necessary—that is, when the model's learning dynamics exhibit meaningful change. As a result, our approach remains both highly responsive and computationally efficient. Our ablation study in Appendix B.2 validates this, demonstrating that our dynamic trigger achieves superior accuracy with significantly lower overhead compared to fixed-interval strategies.

## 3 EXPERIMENTS

### 3.1 EXPERIMENTAL SETUP

**Datasets and models** We evaluate the effectiveness of our proposed method on two representative machine learning tasks: image classification and language modeling. For image classification, we train ResNet-18 (He et al., 2016) on CIFAR-10 (Krizhevsky et al., 2009) using momentum SGD with momentum 0.9, and ViT-Small (Dosovitskiy et al., 2020) on ImageNet (Deng et al., 2009) using AdamW with gradient clipping and weight decay. For language modeling, we employ a two-layer

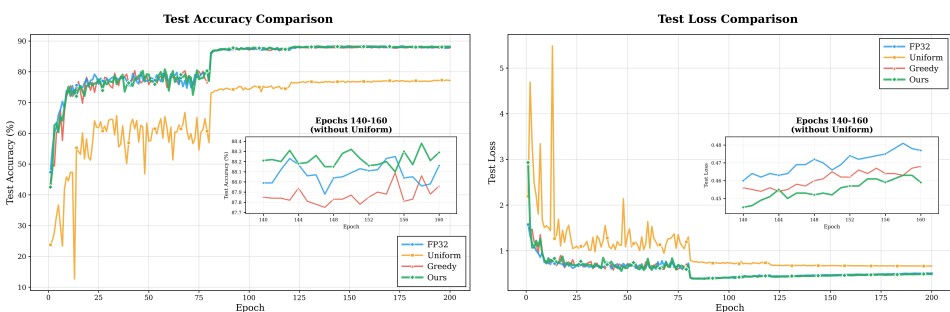

Figure 2: Learning curves of full-precision and different quantization methods for ResNet-18 on CIFAR-10 under 2-bit budget. The zommed-in views highlight test curves during Epochs 140–160.

LSTM (Press & Wolf, 2016) on Penn Treebank (PTB) (Marcinkiewicz, 1994) trained with vanilla SGD and gradient clipping, and a four-layer Transformer (Vaswani et al., 2017) on WikiText-103 (Merity et al., 2016) using AdamW (Loshchilov & Hutter, 2017) with gradient clipping and weight decay. This selection encompasses a broad range of model architectures, including convolutional and recurrent networks as well as modern attention-based designs.

For ViT-Small on ImageNet, we divide the full training process into three representative 50-epoch intervals: 0–50 (early stage), 150–200 (middle stage), and 250–300 (late stage). This setup enables us to evaluate each method's behavior across distinct training dynamics—from rapid parameter updates in early training to stable convergence in later stages—while significantly reducing computational requirement. The middle and late stages are initialized from FP32 checkpoints at epoch 150 and 250, respectively, and finetuned for 50 epochs using quantized gradient for training. This staged evaluation strategy ensures both fairness and efficiency, avoiding the need for full retraining under each configuration.

**Baselines** Given that the optimization objectives and constraint conditions in prior studies (Markov et al., 2024; Yan et al., 2022) on bit allocation for gradient compression differ from ours, as elaborated in the Introduction, we select the following three baselines for comparison: (1) *'FP32'* refers to full-precision training and serves as the benchmark for evaluating training accuracy; (2) *'Uniform'* applies a fixed bit-width uniformly across all gradient layers; (3) *'Greedy'* represents a heuristic-based adaptive bit allocation method. Specifically, the greedy strategy utilizes the same distortion metric as our method but differs in its approach by iteratively allocating bits to the layer with the highest quantization error until the communication budget is fully utilized, without accounting for global optimality. For a fair comparison, all methods are evaluated under the same average bit budget and are implemented using **uniform scalar quantization**.

**Infrastructure** All experiments are implemented in PyTorch 2.3.0 with CUDA 12.1 and cuDNN 8.9.0.2. We use `torchvision` 0.18.0 and `torchtext` 0.18.0 for data processing of image classification and language modeling, respectively. Model training is performed on NVIDIA RTX 4090 GPUs and AMD EPYC 7402 24-core CPUs at 2.8GHz.

## 3.2 EVALUATION ON DIFFERENT MODELS AND DATASETS

We first conduct experiments on a variety of deep learning models and datasets to assess the effectiveness of our bit allocation framework. Table 1 and Table 2 present the comparison results between our method and baseline approaches on the image classification and language modeling tasks, respectively. The results demonstrate that, under the given bit constraint, our proposed layer-wise bit allocation method consistently enhances final model training performance, as measured by accuracy and perplexity. For instance, on ResNet-18, our method achieves 88.39% accuracy, representing improvements of 11.01% and 0.30% over the 'Uniform' and 'Greedy' baselines, respectively. In the case of ViT-small on ImageNet, our approach outperforms the baselines at every training stage, achieving higher accuracy even at more aggressive compression rates, particularly during the initial 50 epochs. For language modeling with LSTM on PTB, our method attains a perplexity of 388.08,

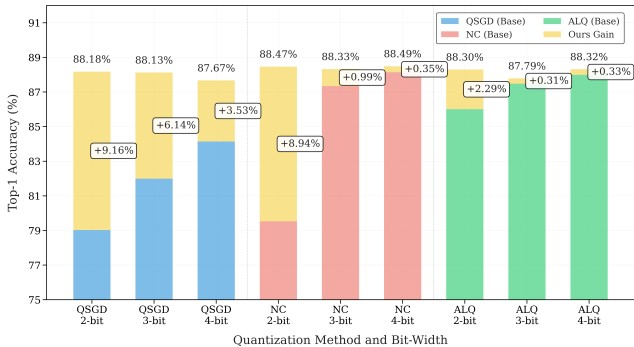

Figure 3: Performance boost from our bit allocation framework across various base quantizers on ResNet-18 using from 2 to 4 bits.

reducing near 40% and 31% perplexity compared to 'Uniform' and 'Greedy' methods, respectively. The advantages of our approach are also evident in Transformer models, where, under a 3-bit quantization setting, it achieves a perplexity of 118.27 at a compression rate of $12.77\times$. These results validate that our bit allocation framework consistently identifies more effective bit allocation strategies than both uniform and heuristic-based adaptive methods across diverse domains and training stages.

### 3.3 COMPARISON ON CONVERGENCE SPEED AND STABILITY

To further investigate the impact of different quantization strategies on training trajectories under constrained communication budgets, we visualize the test loss and accuracy curves. As shown in Figure 2, our method consistently delivers competitive performance throughout the training process, demonstrating clear advantages in both convergence speed and final accuracy. During the early training phase, the 'Uniform' baseline exhibits significant instability and underfitting, as reflected by pronounced fluctuations and elevated test loss. This instability arises from its inability to accommodate layer-wise sensitivity variations under stringent bit constraints. In contrast, both 'Greedy' and our method substantially enhance training stability through adaptive bit allocation. Zoomed-in views of the training curves during the late stage (Epochs 140–160) reveal that while all methods achieve relative stability, our approach attains the highest accuracy and lowest loss with minimal fluctuation, outperforming both 'FP32' and 'Greedy'. These results underscore the efficacy of our bit allocation framework in dynamically adapting to the evolving gradient landscape and effectively prioritizing sensitive layers.

### 3.4 ENHANCED PERFORMANCE WHEN INTEGRATED WITH EXISTING QUANTIZERS

To demonstrate the versatility of our bit allocation framework, we also apply our adaptive bit allocation strategy to several representative base quantizers, including QSGD (Alistarh et al., 2017), NC (Horvóth et al., 2022), and ALQ (Faghri et al., 2020). These methods span a diverse range of quantization paradigms, from fixed uniform schemes (QSGD) to non-uniform quantization (NC) and adaptive techniques (ALQ). As shown in Figure 3, we compare the accuracy of each quantizer with and without the integration of our bit allocation framework under an identical average bit budget. The results demonstrate that our framework consistently achieves significant performance gains across all quantizers, particularly under aggressive low-bit settings. For instance, applying our method to QSGD with a 2-bit budget elevates accuracy from 79.02% to 88.18%, effectively recovering 9.16% in performance and mitigating the adverse effects of severe quantization. Similarly, NC benefits from an accuracy increase from 82.93% to 88.11%, demonstrating that even advanced non-uniform schemes can be further enhanced by our strategy. Even for a strong baseline like ALQ, which already adapts its quantization range, our method further improves its performance from 86.01% to 88.30% at 2-bit setting. By integrating seamlessly with a wide range of existing quantization methods, our approach enables significant improvements in both gradient compression efficiency and final model performance.

## 4 CONCLUSION

In this paper, we proposed an efficient layer-wise bit allocation framework for gradient quantization based on a principled RDO formulation. Departing from prior works that naively optimize for signal-level quantization errors, we introduced a loss-aware distortion measure that captures the sensitivity of training loss to quantization, enabling task-aligned optimization. A key contribution of our work is the discovery and validation—both theoretical and empirical—of the linear superposition property of loss distortion. This property allows us to decompose the otherwise intractable joint bit allocation problem into a series of decoupled, per-layer subproblems. Building on this insight, we developed a Lagrangian-based optimization algorithm that finds the globally optimal bit allocation with linear computational complexity in the number of layers. Extensive experiments across a wide range of model architectures and learning tasks demonstrate the superiority and generality of our approach. Our method consistently outperforms both static and adaptive baselines, and can be seamlessly integrated with various quantization schemes to further enhance model performance under constrained communication budgets. Beyond empirical gains, this work offers theoretical insights into the structure of optimization-aware compression strategies, contributing toward scalable and intelligent gradient quantization for large-scale distributed deep learning.

## 5 ETHICS STATEMENT

This work adheres to the ICLR Code of Ethics. In this study, no human subjects or animal experimentation was involved. All datasets used were sourced in compliance with relevant usage guidelines, ensuring no violation of privacy. We have taken care to avoid any biases or discriminatory outcomes in our research process. No personally identifiable information was used, and no experiments were conducted that could raise privacy or security concerns. We are committed to maintaining transparency and integrity throughout the research process.

## 6 REPRODUCIBILITY STATEMENT

All source code required for conducting and analyzing the experiments **will be made publicly available upon publication of the paper** with a license that allows free usage for research purposes. The experimental setup, including model configurations and hardware details, is described in detail in the main paper (Section 3.1). Additionally, all datasets used in the paper are publicly available, ensuring consistent and reproducible evaluation results.

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

# Appendix

## A    PROOF OF THE LINEAR SUPERPOSITION PROPERTY ON ADAMW

In our main paper, we provided a theoretical justification for the linear superposition property of distortion based on a first-order Taylor expansion, primarily framed around the SGD optimizer for clarity. Here, we extend this analysis to the AdamW optimizer (Loshchilov & Hutter, 2017), which is commonly used in practice. This justifies the application of our rate-distortion optimization framework to modern adaptive optimizers. The notation is kept consistent with the analysis for SGD.

Our proof relies on the same set of standard assumptions outlined in the main text, which we restate here for clarity within the context of the AdamW optimizer.

**Assumption 3** (Smoothness). *The loss function $L(W)$ is $L$-smooth, implying its gradient is $L$-Lipschitz continuous and its Hessian is bounded ($\|\nabla^2 L(W)\| \leq L$).*

**Assumption 4** (Small Perturbation Regime). *The quantization errors $e^{(l)}$ are sufficiently small, such that the resulting perturbations to the moments $(\Delta m, \Delta v)$ and weights $(\delta W)$ are small enough for first-order approximations to be accurate.*

Assumption 3 is required in Section A.5 to justify the first-order Taylor expansion of the loss function itself. The AdamW analysis involves two Taylor expansions: one for the non-linear update rule (Section A.4) and one for the loss function (Section A.5). Assumption 4 ensures the validity of both.

Crucially, the proof for AdamW also leverages the **structural orthogonality** of per-layer quantization errors. This is not an assumption but an intrinsic feature of our framework where error vectors for distinct layers are orthogonal (i.e., $(e^{(i)})^T e^{(j)} = 0$ for $i \neq j$). As shown in Section A.3, this property is essential for the non-linear second-moment update to be decomposed exactly into a sum of per-layer perturbations, by eliminating any cross-term interference.

### A.1    PRELIMINARIES: THE ADAMW UPDATE RULE

At timestep $t$, the AdamW optimizer updates the weights $\mathbf{W}_t$ to $\mathbf{W}_{t+1}$ based on the gradient $\mathbf{g}_t = \nabla \mathcal{L}(\mathbf{W}_t)$. The complete update rule is as follows:

$$\mathbf{m}_t = \beta_1 \mathbf{m}_{t-1} + (1 - \beta_1)\mathbf{g}_t, \tag{17}$$

$$\mathbf{v}_t = \beta_2 \mathbf{v}_{t-1} + (1 - \beta_2)\mathbf{g}_t^2, \tag{18}$$

$$\hat{\mathbf{m}}_t = \frac{\mathbf{m}_t}{1 - \beta_1^t}, \tag{19}$$

$$\hat{\mathbf{v}}_t = \frac{\mathbf{v}_t}{1 - \beta_2^t}, \tag{20}$$

$$\Delta \mathbf{W}_t = \eta \frac{\hat{\mathbf{m}}_t}{\sqrt{\hat{\mathbf{v}}_t} + \epsilon}, \tag{21}$$

$$\mathbf{W}_{t+1} = \mathbf{W}_t - \Delta \mathbf{W}_t - \eta \lambda \mathbf{W}_t, \tag{22}$$

where $\mathbf{m}_t$ and $\mathbf{v}_t$ are the first and second moment estimates, $\hat{\mathbf{m}}_t$ and $\hat{\mathbf{v}}_t$ are their bias-corrected counterparts, and all operations involving vectors are element-wise. For this analysis, we focus on the perturbation to the main update term $\Delta \mathbf{W}_t$, as the decoupled weight decay term is independent of the gradient quantization.

### A.2    DEFINING WEIGHT PERTURBATION DUE TO QUANTIZATION

Our goal is to analyze the relationship between the gradient quantization error and the resulting perturbation in the weight space. Let $\Delta \mathbf{W}_{\text{orig},t}$ denote the weight update computed using the full-precision gradient $\mathbf{g}_t$ and $\Delta \mathbf{W}_{\text{mixed},t}^{(\mathcal{K})}$ denote the update computed using a mixed gradient where all layers in a set $\mathcal{K}$ are quantized. The weight perturbation, $\delta \mathbf{W}_t^{(\mathcal{K})}$, is the difference between these

two updates:

$$\delta \mathbf{W}_t^{(\mathcal{K})} \triangleq \Delta \mathbf{W}_{\text{mixed},t}^{(\mathcal{K})} - \Delta \mathbf{W}_{\text{orig},t}. \tag{23}$$

The key to establishing the property is to determine if $\delta \mathbf{W}_t^{(\mathcal{K})} \approx \sum_{l \in \mathcal{K}} \delta \mathbf{W}_t^{(l)}$, where $\delta \mathbf{W}_t^{(l)}$ is the perturbation from quantizing only layer $l$.

### A.3 ADDITIVITY OF PERTURBATIONS IN MOMENT ESTIMATES

First, we analyze how quantization errors propagate to the moment estimates.

**First Moment Perturbation**  The perturbed first moment estimate, $\mathbf{m}_{\text{mixed},t}^{(\mathcal{K})}$, is:

$$\mathbf{m}_{\text{mixed},t}^{(\mathcal{K})} = \beta_1 \mathbf{m}_{t-1} + (1 - \beta_1) \mathbf{g}_{\text{mixed},t}^{(\mathcal{K})} \tag{24}$$

$$= \beta_1 \mathbf{m}_{t-1} + (1 - \beta_1) \left( \mathbf{g}_t - \sum_{l \in \mathcal{K}} \mathbf{e}_t^{(l)} \right) \tag{25}$$

$$= (\beta_1 \mathbf{m}_{t-1} + (1 - \beta_1) \mathbf{g}_t) - (1 - \beta_1) \sum_{l \in \mathcal{K}} \mathbf{e}_t^{(l)} \tag{26}$$

$$= \mathbf{m}_{\text{orig},t} - (1 - \beta_1) \sum_{l \in \mathcal{K}} \mathbf{e}_t^{(l)}. \tag{27}$$

The total perturbation to the first moment is thus $\Delta \mathbf{m}_t^{(\mathcal{K})} = \mathbf{m}_{\text{mixed},t}^{(\mathcal{K})} - \mathbf{m}_{\text{orig},t} = -(1 - \beta_1) \sum_{l \in \mathcal{K}} \mathbf{e}_t^{(l)}$. Since the single-layer perturbation is $\Delta \mathbf{m}_t^{(l)} = -(1 - \beta_1) \mathbf{e}_t^{(l)}$, we have:

$$\Delta \mathbf{m}_t^{(\mathcal{K})} = \sum_{l \in \mathcal{K}} \Delta \mathbf{m}_t^{(l)}. \tag{28}$$

**Second Moment Perturbation**  The perturbed second moment estimate, $\mathbf{v}_{\text{mixed},t}^{(\mathcal{K})}$, is:

$$\mathbf{v}_{\text{mixed},t}^{(\mathcal{K})} = \beta_2 \mathbf{v}_{t-1} + (1 - \beta_2)(\mathbf{g}_{\text{mixed},t}^{(\mathcal{K})})^2. \tag{29}$$

The total perturbation is $\Delta \mathbf{v}_t^{(\mathcal{K})} = \mathbf{v}_{\text{mixed},t}^{(\mathcal{K})} - \mathbf{v}_{\text{orig},t} = (1 - \beta_2)[(\mathbf{g}_{\text{mixed},t}^{(\mathcal{K})})^2 - (\mathbf{g}_t)^2]$. We expand the squared term:

$$(\mathbf{g}_{\text{mixed},t}^{(\mathcal{K})})^2 = \left( \mathbf{g}_t - \sum_{l \in \mathcal{K}} \mathbf{e}_t^{(l)} \right)^2 = (\mathbf{g}_t)^2 - 2\mathbf{g}_t \left( \sum_{l \in \mathcal{K}} \mathbf{e}_t^{(l)} \right) + \left( \sum_{l \in \mathcal{K}} \mathbf{e}_t^{(l)} \right)^2. \tag{30}$$

Due to the zero-padding, the error vectors for different layers are orthogonal, i.e., $\mathbf{e}_t(i) \cdot \mathbf{e}_t(j) = 0$ for $i \neq j$. This implies that the square of the sum is the sum of the squares. The perturbation becomes:

$$\Delta \mathbf{v}_t^{(\mathcal{K})} = (1 - \beta_2) \left[ -\sum_{l \in \mathcal{K}} (2\mathbf{g}_t^{(l)} \mathbf{e}_t^{(l)}) + \sum_{l \in \mathcal{K}} (\mathbf{e}_t^{(l)})^2 \right] \tag{31}$$

$$= \sum_{l \in \mathcal{K}} (1 - \beta_2) \left[ -2(\mathbf{g}_t^{(l)} \mathbf{e}_t^{(l)}) + (\mathbf{e}_t^{(l)})^2 \right]. \tag{32}$$

The term inside the summation is precisely the single-layer second-moment perturbation, $\Delta \mathbf{v}_t^{(l)}$. Therefore, the additivity is also exact for the second moment:

$$\Delta \mathbf{v}_t^{(\mathcal{K})} = \sum_{l \in \mathcal{K}} \Delta \mathbf{v}_t^{(l)}. \tag{33}$$

The bias correction terms (Equation 19, 20) are scalar multiplications and do not affect the additivity of these perturbations.

## A.4 FROM MOMENT PERTURBATIONS TO WEIGHT PERTURBATION

The weight update $\Delta\mathbf{W}_t$ is a non-linear function $f(\mathbf{m}_t, \mathbf{v}_t)$. To analyze the weight perturbation $\delta\mathbf{W}_t$, we apply a first-order multi-variate Taylor expansion to $f(\cdot, \cdot)$ around the full-precision point $(\mathbf{m}_{\text{orig},t}, \mathbf{v}_{\text{orig},t})$. For simplicity, we absorb the bias correction scalars into $f$:

$$\delta\mathbf{W}_t^{(\mathcal{K})} = f(\mathbf{m}_{\text{orig},t} + \Delta\mathbf{m}_t^{(\mathcal{K})}, \mathbf{v}_{\text{orig},t} + \Delta\mathbf{v}_t^{(\mathcal{K})}) - f(\mathbf{m}_{\text{orig},t}, \mathbf{v}_{\text{orig},t}) \approx \frac{\partial f}{\partial\mathbf{m}}\Delta\mathbf{m}_t^{(\mathcal{K})} + \frac{\partial f}{\partial\mathbf{v}}\Delta\mathbf{v}_t^{(\mathcal{K})}. \tag{34}$$

The partial derivatives are evaluated at the full-precision point. Since the perturbations to the moments are additive, we substitute them into Equation 34:

$$\delta\mathbf{W}_t^{(\mathcal{K})} \approx \frac{\partial f}{\partial\mathbf{m}}\left(\sum_{l\in\mathcal{K}}\Delta\mathbf{m}_t^{(l)}\right) + \frac{\partial f}{\partial\mathbf{v}}\left(\sum_{l\in\mathcal{K}}\Delta\mathbf{v}_t^{(l)}\right) \tag{35}$$

$$= \sum_{l\in\mathcal{K}}\left(\frac{\partial f}{\partial\mathbf{m}}\Delta\mathbf{m}_t^{(l)} + \frac{\partial f}{\partial\mathbf{v}}\Delta\mathbf{v}_t^{(l)}\right). \tag{36}$$

The term inside the summation in Equation 36 is precisely the first-order approximation of the individual weight perturbation $\delta\mathbf{W}_t^{(l)}$ from quantizing only layer $l$. Thus, we have established the approximate additivity of weight perturbations:

$$\delta\mathbf{W}_t^{(\mathcal{K})} \approx \sum_{l\in\mathcal{K}}\delta\mathbf{W}_t^{(l)}. \tag{37}$$

## A.5 FROM WEIGHT PERTURBATION TO LOSS DISTORTION

Finally, we connect the weight perturbation to the overall loss distortion. Let $\mathbf{W}_{\text{orig},t+1} = \mathbf{W}_t - \Delta\mathbf{W}_{\text{orig},t}$ be the weight state after a full-precision update (ignoring weight decay). The state after a quantized update is $\mathbf{W}_{\text{mixed},t+1}^{(\mathcal{K})} = \mathbf{W}_t - \Delta\mathbf{W}_{\text{mixed},t}^{(\mathcal{K})} = \mathbf{W}_{\text{orig},t+1} - \delta\mathbf{W}_t^{(\mathcal{K})}$.

The distortion $D_{(\mathcal{K})}$ is the change in loss between these two final states:

$$D_{(\mathcal{K})} = \mathcal{L}(\mathbf{W}_{\text{mixed},t+1}^{(\mathcal{K})}) - \mathcal{L}(\mathbf{W}_{\text{orig},t+1}) = \mathcal{L}(\mathbf{W}_{\text{orig},t+1} - \delta\mathbf{W}_t^{(\mathcal{K})}) - \mathcal{L}(\mathbf{W}_{\text{orig},t+1}). \tag{38}$$

Applying a first-order Taylor expansion to the loss function $\mathcal{L}(\cdot)$ around $\mathbf{W}_{\text{orig},t+1}$ yields:

$$D_{(\mathcal{K})} \approx \nabla\mathcal{L}(\mathbf{W}_{\text{orig},t+1})^T(-\delta\mathbf{W}_t^{(\mathcal{K})}) = -(\mathbf{g}_{t+1})^T\delta\mathbf{W}_t^{(\mathcal{K})}, \tag{39}$$

where $\mathbf{g}_{t+1} = \nabla\mathcal{L}(\mathbf{W}_{\text{orig},t+1})$ is the gradient evaluated at the updated weights.

By substituting the additivity of weight perturbations from Equation 37 and leveraging the linearity of the dot product, we arrive at the final result:

$$D_{(\mathcal{K})} \approx -(\mathbf{g}_{t+1})^T\left(\sum_{l\in\mathcal{K}}\delta\mathbf{W}_t^{(l)}\right) \tag{40}$$

$$= \sum_{l\in\mathcal{K}}\left(-(\mathbf{g}_{t+1})^T\delta\mathbf{W}_t^{(l)}\right) \tag{41}$$

$$\approx \sum_{l\in\mathcal{K}}D_l. \tag{42}$$

Therefore, the total distortion caused by quantizing multiple layers under the AdamW optimizer can be approximated by the sum of distortions from quantizing each layer individually. The approximation error stems from the higher-order terms in two Taylor expansions: one for the non-linear update rule and another for the loss function itself. This indicates that our rate-distortion framework remains theoretically applicable to adaptive optimizers.

Table 3: Ablation results for different distortion metrics on ResNet-18 at 2 and 3-bit quantization.

| Method | Distortion Metric | Acc. at 2-bit (%) | Acc. at 3-bit (%) |
|--------|-------------------|-------------------|-------------------|
| Uniform | - | 77.33 | 87.32 |
| + Greedy | MSE | 87.13 | 88.14 |
| | LAD (Ours) | **88.09** | **88.33** |
| + Ours | MSE | 87.24 | 88.21 |
| | LAD (Ours) | **88.39** | **88.49** |

Table 4: Ablation results for different bit reallocation strategies and trigger configurations. All methods are evaluated on ResNet-18 using 2-bit quantization. The "Strategy Explanation" column briefly summarizes the dynamic behavior of each setting.

| Strategy Type | Configuration | Accuracy (%) | Total Reallocations | Strategy Explanation |
|---------------|---------------|--------------|---------------------|----------------------|
| Static Baselines | $\tau = 0.00$ | 87.31 | 1 | Only allocates at the first iteration |
| Fixed Interval Baselines | Fixed-50 | 88.02 | 784 | Re-allocates every 50 iterations |
| | Fixed-100 | 88.26 | 392 | Re-allocates every 100 iterations |
| Dynamic Trigger (Ours) Sensitivity to $\tau$ (with $k_{\min} = 20$) | $\tau = 0.98$ | 87.80 | 373 | High sensitivity |
| | $\tau = 0.95$ | **88.32** | 91 | **Balanced trade-off** |
| | $\tau = 0.92$ | 88.28 | 59 | Low sensitivity |
| Dynamic Trigger (Ours) Sensitivity to $k_{\min}$ (with $\tau = 0.95$) | $k_{\min} = 0$ | **88.39** | 416 | No minimum reallocation interval |
| | $k_{\min} = 20$ | 88.32 | 91 | **Balanced trade-off** |
| | $k_{\min} = 50$ | 88.20 | 51 | Longer reallocation interval |

# B  ABLATION STUDIES

## B.1  EFFECTIVENESS OF THE DISTORTION METRIC

We validate the core principle of our approach through an ablation study designed to assess the effectiveness of the proposed LAD metric (Equation 3) compared with traditional signal-level metrics. Specifically, we evaluate both our Lagrangian-based allocation method and the Greedy baseline when guided by either MSE or LAD.

As shown in Table 3, the results provide strong evidence for the superiority of the LAD metric. First, for both allocation algorithms, replacing MSE with LAD consistently improves final accuracy, confirming that directly quantifying the impact on the training objective offers a more reliable signal for bit allocation. Second, we observe a pronounced synergy between our proposed metric and our optimization algorithm: when guided by the suboptimal MSE metric, the performance gap between our Lagrangian method and the greedy heuristic is relatively small, whereas under LAD guidance the advantage of our method becomes substantial. This demonstrates that the full potential of our principled optimization approach is realized when it is paired with its corresponding principled distortion measure.

## B.2  ANALYSIS OF THE DYNAMIC REALLOCATION TRIGGER

We conduct an ablation study to evaluate the effectiveness of our proposed dynamic reallocation trigger and analyze its sensitivity to two key hyperparameters: the similarity threshold $\tau$ and the minimum reallocation interval $k_{\min}$. The parameter $\tau$ governs the sensitivity of the trigger to changes in the gradient distribution. Larger values make the system more responsive by lowering the similarity required to trigger reallocation. Conversely, $k_{\min}$ serves as a damping factor, enforcing a minimum number of iterations between successive reallocations; smaller values permit more frequent updates.

Table 4 presents the performance of different configurations. Notably, disabling reallocation after initialization by setting $\tau = 0$ results in a significant accuracy drop to 87.31%, highlighting the critical role of dynamic reallocation in preserving model performance under quantization constraints.

We also examine fixed-interval reallocation strategies, such as triggering reallocation every 50 or 100 steps. While these strategies partially recover accuracy, they incur substantial computational overhead, requiring up to 784 reallocation events over the course of training. In contrast, dynamic strategies based on our trigger mechanism achieve superior trade-offs. Among dynamic configurations, $\tau = 0.95$ consistently provides a favorable trade-off between accuracy and efficiency. Although the setting with $\tau = 0.95$ and $k_{\min} = 0$ achieves the highest accuracy of 88.39%, increasing $k_{\min}$ to 20 reduces the number of reallocations from 416 to 91, with only a 0.07% decrease in accuracy. These results confirm the necessity of adaptive reallocation and the importance of tuning the trigger mechanism. Our proposed dynamic trigger significantly outperforms fixed or naive strategies, enabling efficient training with minimal accuracy loss under aggressive quantization.

# C  THEORETICAL ANALYSIS OF CONVERGENCE GUARANTEES

In this section, we provide a comprehensive theoretical analysis of of our proposed method. Our analysis is structured to first establish the formal convergence guarantees by proving the key properties of our composite quantizer (Section C.1). We then provide a deeper insight into why our method accelerates optimization by linking our LAD metric to the step-by-step dynamics (Section C.2). Finally, we analyze the robustness of the decoupled RDO formulation (Section C.3).

We begin by formally stating the standard assumptions underpinning our analysis.

**Assumption 5** (L-smoothness). *The loss function $L(W)$ is differentiable and L-smooth, i.e., there exists a constant $L > 0$ such that for all $W_1, W_2$: $\|\nabla L(W_1) - \nabla L(W_2)\| \leq L\|W_1 - W_2\|$.*

**Assumption 6** (Lower Boundedness). *The loss function is lower-bounded by a scalar $L^*$.*

**Assumption 7** (Unbiased Stochastic Gradients with Bounded Variance). *The stochastic gradient $\mathbf{g}_t$ is an unbiased estimator of the true gradient, $\mathbb{E}_{\mathcal{D}}[\mathbf{g}_t|W_t] = \nabla L(W_t)$, and has a bounded variance, $\mathbb{E}_{\mathcal{D}}[\|\mathbf{g}_t - \nabla L(W_t)\|^2|W_t] \leq \sigma_g^2$, where the expectation is over the data sampling.*

**Assumption 8** (Quantizer Properties). *The base quantizers $\mathcal{Q}_b$ for each bit-width $b \in \mathcal{B}_{options}$ satisfy either:*

(a) ***Unbiasedness with Bounded Variance:*** $\mathbb{E}_{\mathcal{Q}}[\mathcal{Q}_b(\mathbf{v})] = \mathbf{v}$ *and* $\mathbb{E}_{\mathcal{Q}}[\|\mathcal{Q}_b(\mathbf{v}) - \mathbf{v}\|^2] \leq \omega_b\|\mathbf{v}\|^2$.

(b) ***Bounded Second Moment:*** $\mathbb{E}_{\mathcal{Q}}[\|\mathcal{Q}_b(\mathbf{v}) - \mathbf{v}\|^2] \leq (1 - \delta_b)\|\mathbf{v}\|^2 + C_b$.

*Here, the expectation $\mathbb{E}_{\mathcal{Q}}$ is over the randomness of the quantizer. In (a), $\omega_b \geq 0$ is the **relative variance factor** introduced by the unbiased quantizer. In (b), $\delta_b \in [0, 1)$ is a **contraction factor** and $C_b \geq 0$ is an **additive error term**, common for biased quantizers.*

## C.1  FORMAL CONVERGENCE GUARANTEES AND RATE CHARACTERIZATION

### C.1.1  PROPERTIES OF THE COMPOSITE QUANTIZER

Let our composite quantizer be $\mathcal{Q}_{\mathcal{A}}(\mathbf{g})$. We prove that it inherits the properties of the base quantizers.

**Theorem 1** (Unbiasedness of the Composite Quantizer). *If the base quantizers satisfy Assumption 8(a), our composite operator $\mathcal{Q}_{\mathcal{A}}$ is also unbiased with respect to the quantization randomness.*

*Proof.* For a fixed gradient $\mathbf{g}$, the bit allocation $\{b_l\}$ is deterministic. The expectation over the quantizer's randomness $\mathcal{Q}$ is:

$$\mathbb{E}_{\mathcal{Q}}[\mathcal{Q}_{\mathcal{A}}(\mathbf{g})] = \mathbb{E}_{\mathcal{Q}}\left[[\mathcal{Q}_{b_1}(\mathbf{g}^{(1)}), \ldots, \mathcal{Q}_{b_L}(\mathbf{g}^{(L)})]\right]$$
$$= [\mathbb{E}_{\mathcal{Q}}[\mathcal{Q}_{b_1}(\mathbf{g}^{(1)})], \ldots, \mathbb{E}_{\mathcal{Q}}[\mathcal{Q}_{b_L}(\mathbf{g}^{(L)})]]$$
$$= [\mathbf{g}^{(1)}, \ldots, \mathbf{g}^{(L)}]$$
$$= \mathbf{g}.$$

Since the expectation is the identity for any fixed $\mathbf{g}$, the operator is unbiased. $\square$

**Theorem 2** (Variance and Second-Moment Bounds). *Let $\{b_l\}$ be any feasible bit allocation.*

1. ***(Bounded Variance for Unbiased Quantizers)*** *Under Assumption 8(a), $\mathbb{E}_{\mathcal{Q}}[\|\mathcal{Q}_{\mathcal{A}}(\mathbf{g}) - \mathbf{g}\|^2] \leq \omega_{\max}\|\mathbf{g}\|^2$, where $\omega_{\max} = \max_{b \in \mathcal{B}_{options}}\{\omega_b\}$.*

2. ***(Bounded Second Moment for Biased Quantizers)*** *Under Assumption 8(b), $\mathbb{E}_{\mathcal{Q}}[\|\mathcal{Q}_{\mathcal{A}}(\mathbf{g}) - \mathbf{g}\|^2] \leq (1 - \delta_{\min})\|\mathbf{g}\|^2 + C_{alloc}$, where $\delta_{\min} = \min_b\{\delta_b\}$ and $C_{alloc} = \sum_{l=1}^{L} C_{b_l}$ for the specific allocation $\{b_l\}$.*

*Proof.* The proof relies on the layer-wise decomposition of the squared error $\mathbf{e}_l = \mathcal{Q}_{b_l}(\mathbf{g}^{(l)}) - \mathbf{g}^{(l)}$.

$$\mathbb{E}_{\mathcal{Q}}[\|\mathcal{Q}_{\mathcal{A}}(\mathbf{g}) - \mathbf{g}\|^2] = \mathbb{E}_{\mathcal{Q}}\left[\|[\mathbf{e}_1, \ldots, \mathbf{e}_L]\|^2\right] = \sum_{l=1}^{L} \mathbb{E}_{\mathcal{Q}}[\|\mathbf{e}_l\|^2]. \tag{43}$$

**For Case 1 (Unbiased):** Since $b_l \in \mathcal{B}_{options}$, the corresponding variance factor $\omega_{b_l}$ must satisfy $\omega_{b_l} \leq \omega_{\max}$. Therefore:

$$\sum_{l=1}^{L} \mathbb{E}_{\mathcal{Q}}[\|\mathbf{e}_l\|^2] \leq \sum_{l=1}^{L} \omega_{b_l}\|\mathbf{g}^{(l)}\|^2 \leq \sum_{l=1}^{L} \omega_{\max}\|\mathbf{g}^{(l)}\|^2 = \omega_{\max}\|\mathbf{g}\|^2. \tag{44}$$

**For Case 2 (Biased):** The total additive error $C_{\mathrm{alloc}}$ is the sum of the $C_{b_l}$ terms for the *actually chosen* bit-widths.

$$\sum_{l=1}^{L} \mathbb{E}_{\mathcal{Q}}[\|\mathbf{e}_l\|^2] \leq \sum_{l=1}^{L} \left((1 - \delta_{b_l})\|\mathbf{g}^{(l)}\|^2 + C_{b_l}\right) \leq (1 - \delta_{\min})\|\mathbf{g}\|^2 + \sum_{l=1}^{L} C_{b_l}. \tag{45}$$

$\square$

### C.1.2 EXPLICIT CONVERGENCE RATE CHARACTERIZATION

**Lemma 1** (Decomposition of Second Moment). *Let $\tilde{\mathbf{g}}_t = \mathcal{Q}_{\mathcal{A}}(\mathbf{g}_t)$. Under Assumptions 7 and 8(a):*

$$\mathbb{E}[\|\tilde{\mathbf{g}}_t\|^2] \leq (1 + \omega_{eff})(\sigma_g^2 + \|\nabla L(W_t)\|^2), \tag{46}$$

*where $\omega_{eff} \leq \omega_{\max}$ is the effective variance factor for the chosen allocation at step $t$.*

*Proof.* $\mathbb{E}[\|\tilde{\mathbf{g}}_t\|^2] = \mathbb{E}_{\mathcal{D}}[\mathbb{E}_{\mathcal{Q}}[\|\mathcal{Q}_{\mathcal{A}}(\mathbf{g}_t)\|^2|\mathbf{g}_t]] \leq (1 + \omega_{eff})\mathbb{E}_{\mathcal{D}}[\|\mathbf{g}_t\|^2]$. From Assumption 7, $\mathbb{E}_{\mathcal{D}}[\|\mathbf{g}_t\|^2] = \sigma_g^2 + \|\nabla L(W_t)\|^2$. Combining these gives the result. $\square$

**Theorem 3** (Convergence Rate for Non-Convex Objectives). *Under Assumptions 1–8(a) (unbiased case), for $\eta \leq 1/(L(1 + \omega_{\max}))$, after $T$ iterations:*

$$\frac{1}{T}\sum_{t=0}^{T-1} \mathbb{E}[\|\nabla L(W_t)\|^2] \leq \frac{2(L(W_0) - L^*)}{\eta T} + \eta L(1 + \omega_{\max})\sigma_g^2. \tag{47}$$

*Proof.* We follow the standard convergence analysis for SGD with noisy gradients (Bottou et al., 2018). By Assumption 1: $\mathbb{E}[L(W_{t+1})] \leq L(W_t) - \eta\langle\nabla L(W_t), \mathbb{E}[\tilde{\mathbf{g}}_t]\rangle + \frac{\eta^2 L}{2}\mathbb{E}[\|\tilde{\mathbf{g}}_t\|^2]$. By unbiasedness, $\mathbb{E}[\tilde{\mathbf{g}}_t] = \nabla L(W_t)$. Substituting Lemma 1:

$$\mathbb{E}[L(W_{t+1})] \leq L(W_t) - \eta\|\nabla L(W_t)\|^2 + \frac{\eta^2 L}{2}(1 + \omega_{\mathrm{eff}})(\sigma_g^2 + \|\nabla L(W_t)\|^2) \tag{48}$$

$$= L(W_t) - \eta\left(1 - \frac{\eta L(1 + \omega_{\mathrm{eff}})}{2}\right)\|\nabla L(W_t)\|^2 + \frac{\eta^2 L(1 + \omega_{\mathrm{eff}})}{2}\sigma_g^2. \tag{49}$$

Choosing $\eta \leq 1/(L(1 + \omega_{\max}))$ ensures $1 - \frac{\eta L(1+\omega_{\mathrm{eff}})}{2} \geq 1/2$. Rearranging:

$$\frac{\eta}{2}\|\nabla L(W_t)\|^2 \leq \mathbb{E}[L(W_t) - L(W_{t+1})] + \frac{\eta^2 L(1 + \omega_{\max})}{2}\sigma_g^2. \tag{50}$$

A telescoping sum over $t = 0, \ldots, T-1$ and division by $T$ yields the result. $\square$

## C.2 IMPACT ON STEP-BY-STEP OPTIMIZATION DYNAMICS

Beyond asymptotic rates of convergence, we now characterize how our method affects the step-by-step optimization trajectory.

**Proposition 1** (The Connection of LAD to Optimization Degradation). *The total LAD metric $\sum_l D_l(b_l)$ provides a principled estimate of the negative impact of quantization on the one-step loss change.*

*Proof.* Recall the definition $D_l(b_l) = \mathbb{E}_{\mathcal{Q}}[L(W_t - \eta \mathbf{g}_{t,\text{mixed}}^{(l)}) - L(W_t - \eta \mathbf{g}_t)]$, where $\mathbf{g}_{t,\text{mixed}}^{(l)}$ is the gradient vector with only layer $l$ quantized. Let $W_{\text{orig}} = W_t - \eta \mathbf{g}_t$ be the updated weights without quantization, and let the quantization error in layer $l$ be $\mathbf{e}^{(l)} = \mathbf{g}_t^{(l)} - \tilde{\mathbf{g}}_t^{(l)}$. The weight perturbation due to quantizing layer $l$ is $\delta W^{(l)} = \eta \mathbf{e}^{(l)}$.

By Taylor expansion around $W_{\text{orig}}$:

$$D_l(b_l) = \mathbb{E}_{\mathcal{Q}}[L(W_{\text{orig}} + \delta W^{(l)}) - L(W_{\text{orig}})] \tag{51}$$

$$\approx \mathbb{E}_{\mathcal{Q}}[\langle \nabla L(W_{\text{orig}}), \delta W^{(l)} \rangle] + \frac{1}{2} \mathbb{E}_{\mathcal{Q}}[(\delta W^{(l)})^T \nabla^2 L(W_{\text{orig}}) \delta W^{(l)}]. \tag{52}$$

Summing over all layers and utilizing the linear superposition property (Equation 4), the total LAD $\sum_l D_l(b_l)$ approximates the joint loss change. Let $\mathbf{e} = \mathbf{g}_t - \tilde{\mathbf{g}}_t = \sum_l \mathbf{e}^{(l)}$ be the total quantization error.

**For unbiased quantizers** (Assumption 8(a)), $\mathbb{E}_{\mathcal{Q}}[\mathbf{e}^{(l)}] = 0$, so the first-order term vanishes in expectation. By second-order Taylor expansion and noting that quantization errors across layers are independent with zero-padded structure:

$$\sum_l D_l(b_l) \approx \frac{\eta^2}{2} \sum_l \mathbb{E}_{\mathcal{Q}}[(\mathbf{e}^{(l)})^T \nabla^2 L(W_{\text{orig}}) \mathbf{e}^{(l)}]. \tag{53}$$

This term reflects the loss increase due to quantization variance, weighted by the local curvature of the loss landscape (captured by the Hessian $\nabla^2 L$).

**For biased quantizers**, the first-order term dominates:

$$\sum_l D_l(b_l) \approx \eta \langle \nabla L(W_{\text{orig}}), \mathbb{E}_{\mathcal{Q}}[\mathbf{e}] \rangle \tag{54}$$

$$= \eta \langle \nabla L(W_t - \eta \mathbf{g}_t), \mathbb{E}_{\mathcal{Q}}[\mathbf{g}_t - \tilde{\mathbf{g}}_t] \rangle. \tag{55}$$

Since $\nabla L(W_t - \eta \mathbf{g}_t) \approx \nabla L(W_t) + O(\eta)$, to first order in $\eta$:

$$\mathbb{E}[\sum_l D_l(b_l)] \approx \eta \langle \nabla L(W_t), \nabla L(W_t) - \mathbb{E}[\tilde{\mathbf{g}}_t] \rangle = \eta D_t, \tag{56}$$

where $D_t$ is precisely the optimization bias.

Therefore, minimizing $\sum_l D_l(b_l)$ is a principled strategy directly aligned with improving the one-step optimization progress: by reducing bias (for biased quantizers) or by mitigating the negative effects of variance (for unbiased quantizers). This provides the theoretical justification for the faster convergence observed in Figure 2. $\square$

## C.3 ANALYSIS OF THE DECOUPLED RDO FORMULATION

A central element of the computational efficiency of our model is the decoupling of the joint RDO problem, enabled by the linear superposition property (Equation 4). A natural question is how the approximation error inherent in this property affects the quality of the obtained solution. Here, we formally analyze the robustness of our decoupled formulation.

**Lemma 2** (Robustness of Decoupled Solution). *Let $\{b_l^*\}$ be the solution to our decoupled problem and $\{b_l^\dagger\}$ be the true solution to the joint problem. If $|D_{joint} - \sum_l D_l| \le \epsilon \cdot D_{joint}$, then our solution is near-optimal:*

$$D_{joint}(\{b_l^*\}) \le \left( \frac{1 + \epsilon}{1 - \epsilon} \right) D_{joint}(\{b_l^\dagger\}) \tag{57}$$

Table 5: Statistical analysis of bit allocation strategies for ResNet-18. Lower values for balance and stability metrics indicate a more balanced and stable allocation strategy.

| Method | Balance (Mean of Std. Dev.) | Stability (Std. Dev. of Std. Dev.) |
|---|---|---|
| Greedy | 2.37 | 0.14 |
| **Ours (Lagrangian)** | **2.26** | **0.12** |

*Proof.* Let $D^* = D_{\text{joint}}(\{b_l^*\})$ and $D^\dagger = D_{\text{joint}}(\{b_l^\dagger\})$. By the optimality of $\{b_l^*\}$ for the decoupled problem: $\sum_l D_l(b_l^*) \leq \sum_l D_l(b_l^\dagger)$. From the approximation error bound $|D_{\text{joint}} - \sum_l D_l| \leq \epsilon \cdot D_{\text{joint}}$, we have:

$$(1 - \epsilon)D_{\text{joint}} \leq \sum_l D_l \leq (1 + \epsilon)D_{\text{joint}}. \tag{58}$$

Applying these bounds to construct a chain of inequalities:

$$(1 - \epsilon)D^* \leq \sum_l D_l(b_l^*) \leq \sum_l D_l(b_l^\dagger) \leq (1 + \epsilon)D^\dagger. \tag{59}$$

From the outer terms: $(1 - \epsilon)D^* \leq (1 + \epsilon)D^\dagger$. Dividing both sides by $(1 - \epsilon)$ (valid for $\epsilon < 1$) yields: $D^* \leq \frac{1+\epsilon}{1-\epsilon}D^\dagger$. For small $\epsilon \ll 1$, using the approximation $(1 - \epsilon)^{-1} = 1 + \epsilon + O(\epsilon^2)$:

$$\frac{1 + \epsilon}{1 - \epsilon} = (1 + \epsilon)(1 + \epsilon + O(\epsilon^2)) \tag{60}$$

$$= 1 + \epsilon + \epsilon + \epsilon^2 + O(\epsilon^2) \tag{61}$$

$$= 1 + 2\epsilon + O(\epsilon^2). \tag{62}$$

Therefore, $D^* \leq (1 + 2\epsilon)D^\dagger + O(\epsilon^2)$, confirming near-optimality. $\square$

## D  DETAILED ANALYSIS OF BIT ALLOCATION DYNAMICS

To provide deeper insight into the behavior of our bit allocation framework, this section visualizes and analyzes the per-layer bit allocations produced by our method in comparison to the 'Greedy' and 'Uniform' baselines.

Figure 4 depicts the allocation strategies for ResNet-18 at early, middle, and late stages of training under a 2-bit average budget. The results show that both our method and 'Greedy' produce highly heterogeneous allocations, assigning high bit-widths (e.g., 8 bits) to sensitive layers like the initial 'conv1' and final 'fc' layers. This confirms the necessity of a layer-wise adaptive strategy. However, our method appears to find a more balanced distribution, frequently utilizing intermediate bit-widths. In contrast, the Greedy approach often makes more extreme decisions, which appear to shift more frequently between training stages.

To objectively validate these visual observations, we measure *balance* via the mean of the per-step standard deviation. A lower value indicates a less extreme, more balanced allocation on average. Similarly, we measure *stability* via the standard deviation of this metric (volatility), where a lower value indicates a more consistent allocation strategy over time. Table 5 shows that our method achieves a lower mean standard deviation of 2.26 than 'Greedy', confirming it produces more globally balanced and less extreme bit allocations. Moreover, our strategy is significantly more stable, as reflected by its lower volatility score. In summary, these results demonstrate that our method yields allocation strategies that are not only different from 'Greedy' but are measurably more balanced and stable, which helps explain the superior final performance and robustness of our approach.

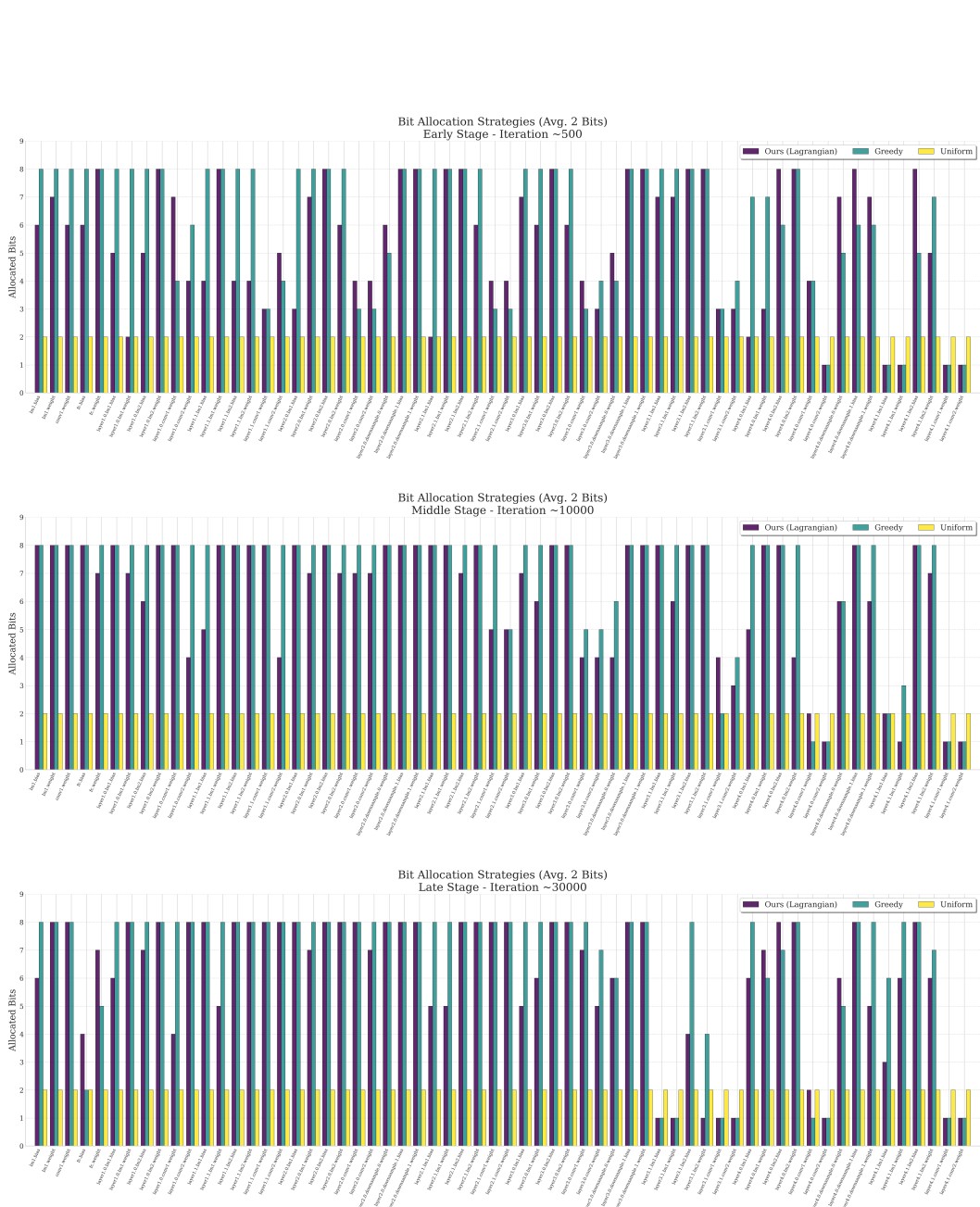

Figure 4: Visualization of per-layer bit allocation strategies, comparing our Lagrangian method, 'Greedy', and 'Uniform' baselines for ResNet-18 trained on CIFAR-10 with a 2-bit average budget. The panels depict the allocations at three distinct training stages: Early (Iteration 500), Middle (Iteration 10,000), and Late (Iteration 30,000).

