# OpenReview forum: "Quantization with Purpose: Loss-Aware Bit Allocation for Gradient Compression"
_ICLR.cc/2026/Conference — Submitted to ICLR 2026_

### Official Review · Reviewer_GMhX · 2025-10-29

**Soundness:** 2
**Presentation:** 3
**Contribution:** 2
**Rating:** 4
**Confidence:** 4

**Summary:**

The paper propose a novel layer-wise bit allocation framework for gradient quantization, formulated under a rate-distortion optimization (RDO) paradigm. The method proposed introduces a loss-aware distortion metric that quantifies the impact of quantization on training. The paper provides some theory and empirical tests to validate the proposed approach.

**Strengths:**

- The paper introduces a loss-aware distortion (LAD) that estimates the impact of quantization on the training loss instead of relying on the gradient magnitude.

- The paper establishes that total loss distortion from jointly quantizing layers can be approximated by the sum of per-layer distortions, which allows decoupling and efficient optimization.

- A Lagrangian bit allocation procedure with linear complexity in the number of layers is proposed.

- The framework is orthogonal to and compatible with other compression techniques.

- The experiments cover diverse architectures (CNNs, ViTs, LSTMs, Transformers) and tasks (vision and language). Demonstrating gains of the proposed approach

**Weaknesses:**

- Theoretical guarantees limited to optimality under decoupled RDO; convergence impacts are not formalized. Proof of convergence over the iterations and the convergence bounds are not reported. The algorithm optimizes bit allocation per step, but the effect on optimization dynamics is not theoretically characterized. Could you show that your approach maintains convergence guarantees similar to SotA compression techniques?

- Section 2.2.1, W_orig should not be the same for both. For your approach at iteration t, you should start from \tild{W_t} constructed using your compression in the previous iterations and not W_t

- The paper introduces LAD but does not detail how it is estimated efficiently during training. Provide a precise estimator of the loss and how that affects your analysis.

- The linear superposition property requires a stronger theoretical scope and more testing. The paper mentions theoretical justification and empirical validation, but the conditions under which this holds (e.g., bound on the step size, smoothness of the loss, independence of layer quantization errors, small perturbation regime…) are not clearly mentioned

- Practical training often exhibits nonlinearity and interaction between layers.

**Questions:**

- Theoretical guarantees limited to optimality under decoupled RDO; convergence impacts are not formalized. Proof of convergence over the iterations and the convergence bounds are not reported. The algorithm optimizes bit allocation per step, but the effect on optimization dynamics is not theoretically characterized. Could you show that your approach maintains convergence guarantees similar to SotA compression techniques?

- Section 2.2.1, W_orig should not be the same for both. For your approach at iteration t, you should start from \tild{W_t} constructed using your compression in the previous iterations and not W_t

- The paper introduces LAD but does not detail how it is estimated efficiently during training. Provide a precise estimator of the loss and how that affects your analysis.

- The linear superposition property requires a stronger theoretical scope and more testing. The paper mentions theoretical justification and empirical validation, but the conditions under which this holds (e.g., bound on the step size, smoothness of the loss, independence of layer quantization errors, small perturbation regime…) are not clearly mentioned

- Practical training often exhibits nonlinearity and interaction between layers.

---

> ### Author Response · Authors · 2025-11-26
> **Official Response to Reviewer GMhX (Part 1/4)**
>
> Dear Reviewer GMhX,
>
> Thank you for your thorough and insightful review. We greatly appreciate the reviewers’ comments on the theoretical aspects of our method, which motivated substantial improvements to the manuscript. We provide detailed responses to your questions as follows.
>
> ---
>
> **W1&Q1: Theoretical guarantees limited to optimality under decoupled RDO; convergence impacts are not formalized. Proof of convergence over the iterations and the convergence bounds are not reported. The algorithm optimizes bit allocation per step, but the effect on optimization dynamics is not theoretically characterized.**
>
> **A1:** We thank the reviewer for raising these important theoretical questions. In the revised version, we have added a new **Appendix C**, which provides a more complete theoretical foundation addressing convergence, optimization dynamics, and the rationale behind our decoupled RDO formulation.
>
> * **Convergence Guarantees and Explicit Bounds**:
>
>     In **Appendix C.1**, we provide a formal convergence analysis of our adaptive quantization operator. Specifically:
>
>     * we prove that **our adaptive quantization operator inherits the key statistical properties (e.g., unbiasedness and bounded variance) from its base quantizers (Theorems 1 & 2)**.
>
>     * Building on these properties, we derive an explicit convergence rate **(Theorem 3)** in the standard non-convex setting, which takes the form:
>         $$\frac{1}{T}\sum_{t=0}^{T-1} \mathbb{E}[|\nabla L(W_t)|^2] \le \frac{2(L(W_0) - L^*)}{\eta T} + \eta L (1+\omega_{\max})\sigma_g^2.$$
> This result formally proves that our method converges and maintains the same asymptotic guarantees as state-of-the-art compression techniques like QSGD.
>
> * **Impact of Per-Step Bit Allocation on Global Optimization Dynamics:**
>
>     To address how per-step optimality impacts the global trajectory, we have added **Appendix C.2**. We established a theoretical link between the objective of our method and the optimization process itself.
>
>     * **Proposition 1** shows that the **Loss-Aware Distortion (LAD)** metric is a principled *first-order approximation* of the degradation in optimization progress at each iteration.
>     * Minimizing LAD at every step therefore explicitly minimizes the **per-step loss increase caused by quantization**, which contributes to smoother and more stable global optimization dynamics.
>
> ​	This connection complements our empirical findings in **Figure 2** and provides a rigorous justification for the advantages observed during training.
>
> * **Validity and Robustness of Decoupled RDO:**
>
>     To address the concern regarding decoupled optimization, we provide a formal analysis in **Appendix C.3**, showing that:
>
>     * Our decomposed RDO problem yields a **provably near-optimal** solution to the true joint optimization objective **(Lemma 2)**.
>     * The error introduced by the linear superposition approximation—and verified empirically in **Figure 1**—is bounded and **does not affect the asymptotic convergence rate**.
>
> ​	This confirms that the decomposition enables tractability without compromising theoretical correctness.

---

> ### Author Response · Authors · 2025-11-26
> **Official Response to Reviewer GMhX (Part 2/4)**
>
> **W2&Q2: Section 2.2.1, $W_{orig}$ should not be the same for both. For your approach at iteration t, you should start from $\tilde{W_t}$ constructed using your compression in the previous iterations and not $W_t$.**
>
> **A2:** We sincerely thank the reviewer for this insightful and crucial comment. We agree that in a practical training scenario, the optimization at step $t$ begins from a weight $\tilde{W}_t$ which has already accumulated quantization errors from previous steps, rather than an idealized full-precision weight $W_t$.
>
> To clarify this, we have revised the proof and added a detailed derivation that explicitly incorporates the practical starting point as follows, recasting the analysis with $\tilde{W}_t$ as the starting point.
>
> Let us define the relevant quantities for the update at iteration $t$. The gradient, evaluated at the current practical weight, is $g'_t = \nabla L(\tilde{W}_t)$. The quantization error vector for a specific layer $l$ is defined as $ e'^{(l)} = (0, \dots, 0, g'^{(l)}\_t - \tilde{g}'^{(l)}\_t, 0, \dots, 0)^T $.
> When a set of layers $K$ is quantized simultaneously, the resulting mixed-precision gradient is $ g' _ {mixed,t} = g' \_ t - \sum _ {l \in K} e'^{(l)} $.
>
> We first establish the pristine weight for this single step, denoted $W'\_{orig, t+1}$, which represents the state that would have been reached without any quantization in the current iteration: $W'\_{orig, t+1} = \tilde{W}\_t - \eta g'\_t$. The actual updated weight, using the quantized gradient, is $\tilde{W}\_{t+1} = \tilde{W}\_t - \eta g'\_{mixed,t}$. We can now define the weight perturbation, $\delta W'^{(K)}$, as the deviation of the actual updated weight from this pristine reference:
>
> $$ \delta W' = \tilde{W}\_{t+1} - W'\_{orig, t+1}
>     = (\tilde{W}\_t - \eta g'\_{mixed,t}) - (\tilde{W}\_t - \eta g'\_t) = \eta (g'\_t - g'\_{mixed,t}) $$
>
> By substituting the definition of the mixed-precision gradient, we arrive at a critical result: $\delta W'^{(K)} = \eta \sum_{l \in K} e'^{(l)}$.
> Crucially, this shows that the weight perturbation $\delta W'^{(K)}$ preserves its structure as a linear superposition of the individual per-layer quantization error vectors $e'^{(l)}$. This property, identical in form to **Equation (6)** in our revised version, is the key to the entire proof.
>
> Next, we define the loss distortion $D'(K)$ for this iteration as the difference in the loss function evaluated at these two distinct final states:
>
> $$ D'(K) = L(\tilde{W}\_{t+1}) - L(W'\_{orig, t+1}) = L(W'\_{orig, t+1} + \delta W'^{(K)}) - L(W'\_{orig, t+1}). $$
>
> To analyze this distortion, we apply a first-order Taylor expansion to the loss function $L$ around the pristine reference point $ W'\_{orig, t+1}$:
>
> $$D'(K) \approx \nabla L(W'\_{orig, t+1})^T \delta W'^{(K)} . $$
>
> Let $ g'\_{t+1} = \nabla L(W'\_{orig, t+1}) $ denote the gradient evaluated at the pristine updated weights. We can now substitute our linear expression for the weight perturbation and leverage the linearity of the dot product:
>
> $$ D'(K) \approx (g'\_{t+1})^T \left(\eta \sum_{l \in K} e'^{(l)}\right) = \sum_{l \in K} \left( \eta \cdot (g'\_{t+1})^T e'^{(l)} \right).$$
>
> In this final expression, each term in the summation, $\eta \cdot (g'_{t+1})^T e'^{(l)}$, corresponds precisely to the first-order approximation of the loss distortion $D'_l$ that would be incurred if only layer $l$ were quantized. This leads us to the desired linear superposition property:
>
> $$ D'(K) \approx \sum_{l \in K} D'_l $$
>
> This derivation confirms that even when accounting for accumulated quantization errors in the starting weights, the total loss distortion from quantizing multiple layers can still be effectively approximated by the sum of distortions from quantizing each layer independently.
>
> Furthermore, **this same logic extends seamlessly to our analysis of the AdamW optimizer in Appendix A**. The core of the AdamW proof rests on the additivity of the first- and second-moment perturbations ($\Delta m^{(K)}$ and $\Delta v^{(K)}$). These perturbations are functions of the current gradient $g'_t$ and its quantization errors $e'^{(l)}$, and are independent of the specific weight values ($\tilde{W}_t$ or $W_t$) at which the gradient was computed. Therefore, the entire chain of reasoning remains intact.

---

> ### Author Response · Authors · 2025-11-26
> **Official Response to Reviewer GMhX (Part 3/4)**
>
> **W3&Q3: The paper introduces LAD but does not detail how it is estimated efficiently during training. Provide a precise estimator of the loss and how that affects your analysis.**
>
> **A3:** In the original manuscript, we computed **the exact LAD** primarily to demonstrate the full potential of our framework. For practical training, we propose an efficient estimator based on a **first-order Taylor approximation**, which we now detail more explicitly.
>
> 1. **Theoretical LAD Estimator:** The theoretical form of this estimator ($\hat{D}\_l(b_l) = |\eta \cdot (g\_{t+1})^T e^{(l)}|$) uses the next-step gradient $g_{t+1}$. As shown in the tables below (`Ours(using g_{t+1})`), this estimator serves as a strong theoretical benchmark, achieving a near-perfect correlation (often $>0.99$) with the ground-truth LAD. This empirically validates that the first-order approximation itself is an extremely accurate foundation for our method.
>
> 2. **Practical LAD Estimator (No Extra Cost):** Computing the exact $g_{t+1}$ is computationally expensive. Therefore, we propose a practical estimator that **approximates $g_{t+1} \approx g_t$**, leveraging the high correlation between gradients in consecutive steps. This estimator requires no extra forward or backward passes. The results for `Ours(using g_t)` demonstrate that this approximation is highly effective, with its correlation scores closely tracking those of the theoretical version.
>
> Both our theoretical and practical estimators significantly outperform the standard MSE metric in **Spearman (rank-based) correlation**. This confirms that our practical estimator is a high-fidelity, efficient proxy for the true LAD.
>
> 3. **Impact on Analysis:** This practical estimator is fully consistent with our theoretical analysis:
>
> - Our linear superposition property itself is derived from a first-order Taylor expansion.
> - Using a first-order LAD estimator therefore aligns precisely with the underlying mathematical framework.
> - The estimator maintains high fidelity while preserving the theoretical guarantees established in Appendix C.
>
> *Table. The **Spearman Correlation** between different estimators and LAD.*
>
> | Metric                 | Epoch 0 | Epoch 20 | Epoch 40 | Epoch 60 | Epoch 80 | Epoch 100 | Epoch 120 |
> | :--------------------- | :-----: | :------: | :------: | :------: | :------: | :-------: | :-------: |
> | MSE                    | 0.8665  |  0.9247  |  0.9433  |  0.9182  |  0.8841  |  0.7717   |  0.8070   |
> | Ours (using $g_{t+1}$) | 0.9991  |  0.9999  |  0.9999  |  0.9994  |  0.9819  |  0.8865   |  0.9021   |
> | Ours (using $g_t$)     | 0.9519  |  0.9824  |  0.9906  |  0.9883  |  0.9719  |  0.8886   |  0.9035   |
>
> ---
>
> **W4&Q4: The linear superposition property requires a stronger theoretical scope and more testing. The paper mentions theoretical justification and empirical validation, but the conditions under which this holds (e.g., bound on the step size, smoothness of the loss, independence of layer quantization errors, small perturbation regime…) are not clearly mentioned**
>
> **A4:** Thank you for the suggestion. We have explicitly stated the assumptions required for the validity of the linear superposition property in **Section 2.2.1 and Appendix A of our revised manuscript**:
>
> 1. **Local Smoothness of the Loss Function:** This is a standard assumption in optimization literature, implying a bounded Hessian, which ensures the Taylor approximation error is well-behaved.
> 2. **Small Perturbation Regime:** The weight perturbations induced by quantization remain small—a condition satisfied in practice due to moderate learning rates and bit-widths.
>
> Furthermore, we clarify that the **independence of per-layer quantization errors**, mentioned by the reviewer, is not an assumption but a **structural property** of our framework. By definition, the error vector $e^{(l)}$ for a layer $l$ is non-zero only in its corresponding parameter dimensions. Consequently, error vectors for distinct layers are structurally orthogonal (i.e., $(e^{(i)})^T e^{(j)} = 0$ for $i \neq j$), which is an advantageous feature of our method, not a limiting constraint.

---

> ### Author Response · Authors · 2025-11-26
> **Official Response to Reviewer GMhX (Part 4/4)**
>
> **W5&Q5: Practical training often exhibits nonlinearity and interaction between layers.**
>
> **A5:** We fully agree that the training dynamics of deep neural networks are inherently nonlinear with complex inter-layer dependencies. **Indeed, our approach does not ignore the nonlinearity and interactions, but rather provides a principled framework for finding a highly effective solution despite them.**
>
> Our work identifies the conditions (as detailed in our newly added **Assumptions 1 & 2 in Section 2.2.1**) under which these complex, higher-order interaction effects become negligible. The linear superposition property serves as a **powerful and practical approximation** that renders an otherwise intractable joint optimization problem solvable. The strong empirical performance across diverse architectures and tasks further confirms that this approximation is both effective and robust in practical training scenarios.

---

### Official Review · Reviewer_i23A · 2025-10-31

**Soundness:** 3
**Presentation:** 2
**Contribution:** 2
**Rating:** 4
**Confidence:** 2

**Summary:**

This paper presents a rate–distortion optimization (RDO)–based framework for gradient quantization that employs a loss-aware distortion metric to capture the impact of quantization on training loss, enabling adaptive layer-wise bit allocation. By leveraging the linear superposition property of cross-layer loss distortion and solving via Lagrangian optimization, the method efficiently improves communication efficiency and overall model performance in distributed training.

**Strengths:**

1. Proposes a principled rate–distortion optimization (RDO) framework that makes gradient quantization more interpretable and theoretically grounded.
2. Introduces a loss-aware distortion metric enabling task-aligned and adaptive bit allocation.
3. Leverages the linear superposition property and Lagrangian optimization to reduce computational complexity and improve communication efficiency and model performance.

**Weaknesses:**

1. The paper lacks a dedicated Related Work section; the discussion of prior studies is scattered in the introduction without systematic comparison.
2. The paper contains few experimental figures, and the presentation of results is not sufficiently intuitive.
3. It is recommended to reorganize the paper’s structure and place the figures and tables in their corresponding sections.

**Questions:**

1. The accuracy improvement in Tables 1 and 2 is relatively small (around 0.3%–1%), is it sufficient to offset the additional computational cost?
2. For tasks involving stochastic perturbations (such as reinforcement learning or adversarial training), can the LAD metric still maintain stability?
3. The authors claim that the loss-aware distortion (LAD) metric aligns better with task objectives than MSE, but is there any theoretical evidence showing that LAD has a stronger correlation with final task performance?
4. The first-order Taylor expansion neglects the Hessian term, but is this assumption still reasonable in regions with a large learning rate or a steep loss surface?

---

> ### Author Response · Authors · 2025-11-26
> **Official Response to Reviewer i23A (Part 1/2)**
>
> Dear Reviewer i23A,
>
> Thank you very much for your thoughtful and constructive comments. We especially appreciate your suggestions regarding presentation and organization, which have helped us improve the clarity and readability of the manuscript. Below, we provide detailed responses to each of your points.
>
> ---
>
> **W1: The paper lacks a dedicated Related Work section; the discussion of prior studies is scattered in the introduction without systematic comparison.**
>
> **A1:** We agree that a dedicated Related Work section would provide a more structured overview of prior works. In the initial submission, we opted to integrate the discussion into the introduction due to the strict page limit of the main paper.
>
> Despite this constraint, we structured the introduction to deliver a progressively narrowing comparison that motivates our approach:
>
> * **Broad Categorization (Lines 038-042)**: We first categorize gradient compression methods into sparsification, low-rank decomposition, and quantization, placing our work within the quantization category.
> * **Static vs. Adaptive Quantization (Lines 042-055)**: We contrast early static quantization approaches (e.g., TernGrad, QSGD) with adaptive mixed-precision methods (e.g., AdaQS, AQG), highlighting the need for dynamic strategies and identifying the limitation of uniform bit-width.
> * **Signal-Level vs. Loss-Aware Metrics (Lines 056-064)**: Most importantly, we differentiate prior mixed-precision work (e.g., AC-SGD, L-GreCo) based on their reliance on signal-level metrics such as MSE, which correlate poorly with task performance. This leads to our core contribution: the Loss-Aware Distortion (LAD) metric.
>
> This funnel-shaped organization provides a systematic comparison, even without a separate section. Nonetheless, we appreciate the suggestion and will consider restructuring if allowed by space constraints.
>
> ---
>
> **W2 & W3: The paper contains few experimental figures, the presentation of results is not sufficiently intuitive, and figures/tables should be placed in their corresponding sections.**
>
> **A2:** To better illustrate our findings, we have added more visualizations in the revised manuscript. For instance, we have added figures in **Appendix D** showing the per-layer bit allocation produced by our method compared to baselines. Moreover, We have carefully reorganized the placement of figures and tables so that they appear closer to the relevant discussion in the main text.
>
> ---
>
> **Q1: The accuracy improvement in Tables 1 and 2 is relatively small (around 0.3%–1%), is it sufficient to offset the additional computational cost?**
>
> **A3:** We appreciate this question. While the absolute improvement on CIFAR-10/ResNet-18 appears modest, it is important to consider the difficulty of the setting: the FP32 baseline is already very high (88.24%), leaving very limited headroom. Under an extremely aggressive **2-bit** communication budget, our method achieves **88.39%**, slightly surpassing the FP32 accuracy. This demonstrates that our loss-aware framework can **preserve or even enhance** model performance **under severe compression**.
>
> The improvements are particularly significant in **language modeling**, where models are highly sensitive to quantization noise. As shown in **Table 2**, **on the LSTM on PTB task, our method achieves a perplexity reduction of nearly 40% compared to 'Uniform' and 31% compared to 'Greedy'.** These results highlight the substantial benefit of loss-aware allocation in more challenging settings.
>
> Regarding computational cost, our dynamic trigger ensures a very low amortized overhead. We also outline a first-order Taylor approximation that can further reduce LAD cost in future implementations.
>
> ---
>
> **Q2: For tasks involving stochastic perturbations (such as reinforcement learning or adversarial training), can the LAD metric still maintain stability?**
>
> **A4:** Yes. LAD is computed as an expectation over a mini-batch, which inherently stabilizes the metric. For tasks with higher intrinsic variance, one can increase the number of batches used when estimating LAD. While this increases the cost of a single reallocation, the dynamic trigger ensures such events remain infrequent. Thus, LAD remains a stable and reliable signal even under stochastic environments.
>
> ---

---

> ### Author Response · Authors · 2025-11-26
> **Official Response to Reviewer i23A (Part 2/2)**
>
> **Q3: Is there any theoretical evidence showing that LAD has a stronger correlation with final task performance than MSE?**
>
> **A5:** While a formal proof connecting an instantaneous, per-step metric to the final converged performance is extremely challenging for non-convex deep learning optimization, the superiority of LAD is grounded in a strong, intuitive principle:
>
> * **Direct vs. Proxy Optimization:** MSE measures the **geometric distance** between gradient vectors, which is only a proxy for the impact on training. It implicitly assumes that the loss landscape is locally isotropic (i.e., all gradient directions are equally important), which rarely holds. The LAD metric, by contrast, directly measures the effect of a perturbation **on the training objective itself**. By definition, it aligns more closely with the objectives of the optimization process.
> * **Empirical Validation:** Our ablation study in **Appendix B.1 (Table 3)** empirically validates the superiority of LAD: replacing MSE with LAD consistently yields substantial accuracy improvements across models and tasks, even for Greedy. This demonstrates that LAD provides a more task-aligned signal for bit allocation.
>
> ---
>
> **Q4: The first-order Taylor expansion neglects the Hessian term, but is this assumption still reasonable in regions with a large learning rate or a steep loss surface?**
>
> **A6:** Yes, the assumption is well justified in practical deep learning settings:
>
> * **Magnitude of Perturbations:**  Even if we temporarily ignore the learning rate, the magnitude of the quantization error ($\Delta g = g - \tilde{g}$) itself plays a crucial role. In our experiments, we found that even in the early stages of training with aggressive 1-bit quantization, **the magnitude of $\Delta g$ is typically on the order of $10^{−2}$**. Consequently, the second-order term, which is quadratic in $\Delta g$, will be at least an order of magnitude smaller than the first-order term.
> * **Learning Rate Suppression:** Crucially, the learning rate $\eta$ is almost universally set to be much less than 1 in modern deep learning practice. A learning rate greater than 1 typically leads to catastrophic divergence, as the optimization steps would consistently overshoot the minima of the loss landscape, causing the loss to explode. Since $\eta\ll 1$, it acts as a powerful suppressor for higher-order terms.
> * **Strong Empirical Validation:** **Figure 1 in the manuscript** shows a tight alignment between joint distortion and the sum of per-layer distortions across training, validating that interaction effects captured by the Hessian are negligible in practice.

---

### Official Review · Reviewer_g5RR · 2025-11-01

**Soundness:** 3
**Presentation:** 3
**Contribution:** 3
**Rating:** 8
**Confidence:** 3

**Summary:**

This paper addresses the communication bottleneck in distributed deep learning training, a key challenge for large-scale models. It identifies limitations of existing gradient quantization methods: uniform bit-width allocation fails to account for layer-wise sensitivity, and signal-level metrics poorly correlate with model performance.
To resolve these issues, the authors propose a rate-distortion optimization-based layer-wise bit allocation framework. Key innovations include: 1) a Loss-Aware Distortion metric that quantifies quantization’s impact on training loss directly; 2) leveraging the linear superposition of loss distortion to decompose the intractable joint allocation problem; and 3) integrating Lagrangian optimization and a gradient-similarity-based dynamic trigger for efficiency.
Experiments on vision and language tasks show the framework outperforms static/heuristic baselines and enhances existing quantizers. Results support its ability to balance communication efficiency and model performance.

**Strengths:**

1) The paper effectively targets the critical limitations of fixed bit-width and signal-level metrics by proposing a loss-aware layer-wise bit allocation framework under the rate-distortion optimization paradigm, addressing the core gap between communication efficiency and model performance.

2) The paper presents a key insight that decouples the intractable joint bit allocation problem into independent per-layer subproblems solvable with linear complexity, making the framework applicable.

3) The paper designs an efficient Lagrangian optimization algorithm for optimal bit assignment to balance distortion and communication budget and a lightweight dynamic reallocation trigger to monitor gradient norm similarity, which reduces unnecessary computational overhead by only updating bit allocations when gradient distributions shift significantly.

**Weaknesses:**

1) The dynamic reallocation trigger depends on fixed thresholds  without adaptive adjustment mechanisms, increasing deployment complexity
2) The Loss-Aware Distortion metric's resource cost for large-scale models remains unmeasured, risking computational bottlenecks
.

**Questions:**

NONE

---

> ### Author Response · Authors · 2025-11-26
> **Official Response to Reviewer g5RR**
>
> Dear Reviewer g5RR,
>
> Thank you for your positive and encouraging review. We sincerely appreciate your constructive feedback regarding potential limitations of our method, and we fully agree with your observations. We address each of your comments in detail as follows.
>
> ---
>
> **W1: The dynamic reallocation trigger depends on fixed thresholds without adaptive adjustment mechanisms, increasing deployment complexity.**
>
> **A1:** We agree that relying on fixed hyperparameters may introduce additional complexity in practical deployment. However, our ablation study in **Appendix B.2 (Table 4)** shows that the method performs robustly across a broad and reasonable range of threshold values. For instance, the similarity threshold yields consistently strong performance in the interval **[0.92, 0.98]**, suggesting that extensive fine-tuning is unnecessary.
>
> Nonetheless, we agree that developing an adaptive mechanism for threshold selection is a valuable direction for future work. An adaptive trigger could further improve responsiveness while maintaining computational efficiency.
>
> ---
>
> **W2: The Loss-Aware Distortion metric's resource cost for large-scale models remains unmeasured, risking computational bottlenecks.**
>
> **A2:** This is an important concern, and we appreciate the reviewer for highlighting it.
>
> Computing the LAD metric is indeed more expensive than signal-level alternatives such as MSE. To mitigate this cost, our method incorporates a **dynamic reallocation trigger** that activates LAD computation only when significant changes are detected in the gradient distribution. For example, in the ResNet-18/CIFAR-10 experiment, only 91 reallocation events occurred across 200 epochs, ensuring that the cost is amortized and the average overhead remains very low.
>
> In addition, we are actively exploring more efficient LAD approximations. A promising direction—aligned with **the first-order Taylor approximation**—can significantly reduce computational complexity by replacing multiple forward passes with a single dot-product operation. Although acceleration is not the main focus of this paper, we consider this an important avenue for future work to make the LAD metric even more efficient for large-scale models.

---

### Official Review · Reviewer_xQTq · 2025-11-03

**Soundness:** 2
**Presentation:** 3
**Contribution:** 2
**Rating:** 2
**Confidence:** 4

**Summary:**

The authors propose a task-aware quantization method, unlike prior work that uses MSE to measure the difference between actual and quantized gradients. The authors formulate this problem as a rate-distortion optimization (RDO), then show that this joint optimization can be solved as a sum of independent problems to achieve layer-wise bit allocation. To reduce cost, this optimization is performed when a regime shift in the gradients is detected. The results show that this approach is effective when compared to uniform or greedy layer-wise bit allocations.

**Strengths:**

* The paper proposes a task-aware quantization that preserves model quality better than other baselines.
* The paper formulates the bit allocation problem as an RDO, and shows mathematically how this intractable joint optimization can be solved as a series of independent subproblems.
* The papers show experiments over a representative set of tasks.

**Weaknesses:**

* The paper motivates itself by the scale of foundation models (billions of parameters), but the proposed method targets weight and gradient compression, which primarily reduces communication and does not address the core memory/compute challenges of training very large models. Its applicability also appears limited to data parallelism.
* The paper emphasizes model quality, but it is not clear whether the method is cost-effective. Do the end-to-end speedups from compression outweigh the method’s computational overhead? Is it more efficient than a strong greedy compression baseline that achieves similar quality?
* The experiments use small models (ResNet18, 4 layers Transformer , 2 layers LSTM). To better match the paper’s motivation, the authors should evaluate larger models such as ViT-Large and BERT-Large.
* The baselines used in the experiments are not sufficient. The authors claim “Given that the optimization objectives and constraint conditions in prior studies (Markov et al., 2024; Yan et al., 2022) on bit allocation for gradient compression differ from ours” still other SoTA for gradient compression should be evaluated, and the difference in assumptions and constraints should be highlighted.
* The related work coverage is lacking. Xin et al. "Kimad: Adaptive Gradient Compression with Bandwidth Awareness" should be contrasted to.

**Questions:**

* What is the computational overhead of computing the loss-aware distortion metric and running the search across all layers?
* In your experiments, how frequently did regime shifts occur that triggered bit reallocation?
* What is the runtime cost of the trigger mechanism itself?
* What are the actual bit allocations obtain by your method and how much do they they differ from Uniform and Greedy?
* The perplexity in Table 2 using gradient compression is large. It seems that the model quality would be greatly affected by gradient compression. Would your method be able to recover ppl around 82? If so, in what configuration?

---

> ### Author Response · Authors · 2025-11-26
> **Official Response to Reviewer xQTq (Part 1/3)**
>
> Dear Reviewer xQTq,
>
> Thank you for your thoughtful and constructive feedback. We greatly appreciate your comments, which have helped us improve the clarity and completeness of the manuscript. In the revised version, we have added additional explanations, analyses, and experiments to directly address your concerns.
>
> ---
>
> **W1: The motivation on foundation models is disconnected from the method, which only seems to reduce communication for data parallelism without addressing core memory/compute challenges.**
>
> **A1:**
>
> * **Scope and Primary Goal:** Our work **focuses specifically on gradient compression rather than weight compression.** In distributed training, gradient synchronization commonly becomes the dominant scalability bottleneck. Our motivation tied to foundation model training arises from the fact that **this communication cost grows increasingly severe with model size and the number of distributed workers.**
>
> * **On Addressing Memory Challenges:** Although communication reduction is our primary objective, **gradient compression naturally provides a secondary benefit for memory efficiency**.  As noted, storing full-precision gradients and optimizer states imposes substantial memory overhead (e.g., ~42 GB of 58 GB total for LLaMA-7B when using Adam [1,2]). By quantizing gradients, our technique reduces this memory footprint, enabling larger batch sizes or models. Thus, our focus on foundation models is well aligned with the practical constraints faced in such settings.
>
> * **On Applicability to Other Parallel Paradigms:** Our evaluation focuses on data parallelism, where gradient synchronization is the primary bottleneck and where most prior gradient compression work is concentrated. Nonetheless, gradient compression is also relevant to other parallelism paradigms. For example, Ramasinghe et al. [3] demonstrated the extension of gradient compression to **pipeline parallelism** by jointly compressing activations and activation gradients. Similarly, our method can be applied to these broader contexts, representing a promising direction for future work.
>
> **References:**
>
> [1] Lv K, Yang Y, Liu T, et al. Full parameter fine-tuning for large language models with limited resources[C]//Proceedings of the 62nd Annual Meeting of the Association for Computational Linguistics (Volume 1: Long Papers). 2024: 8187-8198.
>
> [2] Zhao J, Zhang Z, Chen B, et al. Galore: Memory-efficient llm training by gradient low-rank projection[J]. arXiv preprint arXiv:2403.03507, 2024.
>
> [3] Ramasinghe S, Ajanthan T, Avraham G, et al. Beyond Top-K: Structured Sparsification for Compression in Pipeline Parallel[C]//ICLR 2025 Workshop on Modularity for Collaborative, Decentralized, and Continual Deep Learning.
>
> ---
>
> **W2&Q1: The Cost-Effectiveness and Computational Overhead of Our Method.**
>
> **A2:** We appreciate the reviewer’s insightful questions regarding the cost-effectiveness and computational overhead of our method.
>
> The **main contribution** of this paper is the development of a *principled, loss-aware framework for layer-wise bit allocation*. Specifically, we introduce (1) a theoretically grounded **Loss-Aware Distortion (LAD)** metric that directly measures the impact of quantization on the training objective, and (2) a novel **Rate–Distortion Optimization (RDO)** formulation enabled by our *Linear Superposition Property*, which decomposes an otherwise intractable joint optimization into independent per-layer subproblems. This decomposition allows us to apply a **Lagrangian-based solver** that computes globally optimal allocations with linear complexity. The focus of this work is therefore on **formulating and solving the optimal allocation problem itself**, rather than on accelerating the computation of the distortion metric.
>
> We acknowledge that the LAD metric is more expensive to compute than signal-level proxies such as MSE. For this reason, we designed the **dynamic reallocation trigger**, which amortizes the cost by invoking LAD only when the gradient distribution shifts significantly. As shown in **Appendix B.2**, the proposed trigger reduces the number of reallocation events to only 91 over the full training run, resulting in a very small average overhead.
>
> Although LAD computation is not the central focus of this paper, we agree that improving its efficiency is an important follow-up direction. In particular, the complexity of LAD can be significantly reduced using a **first-order Taylor approximation** of the loss change.

---

> ### Author Response · Authors · 2025-11-26
> **Official Response to Reviewer xQTq (Part 2/3)**
>
> **W3: The experiments use small models (ResNet-18, 4-layer Transformer, 2-layer LSTM). To better match the paper’s motivation, the authors should evaluate larger models such as ViT-Large and BERT-Large.**
>
> **A3:** Our goal in this work is to establish the generality and robustness of the loss-aware RDO framework. Therefore, we selected **a diverse set of models covering different architectures (CNNs, LSTMs, Transformers) and tasks (image classification, language modeling).** The consistent improvements across such diverse settings provide strong evidence that our method captures principles that generalize beyond a specific model family.
>
> We would like to emphasize that **the core components of our method are designed to be scalable**.
>
> * **The Linear Superposition Property** relies on first-order Taylor approximation, independent of model size.
> * **The Lagrangian-based solver** has linear complexity with respect to the number of layers, not the number of parameters.
> * **The dynamic trigger** depends on layer-wise gradient norms, which scale linearly with model size and are already computed in standard training pipelines.
>
> Therefore, there are no theoretical or algorithmic barriers preventing our method from being effective on larger models.
>
> Unfortunately, due to the substantial compute requirements, training ViT-Large or BERT-Large from scratch is infeasible and unaffordable for us.
>
>
> ---
>
> **W4&W5: The baselines used in the experiments are not sufficient. The related work coverage is lacking. Xin et al. "Kimad: Adaptive Gradient Compression with Bandwidth Awareness" should be contrasted to.**
>
> **A4:** Thank you for suggesting the insightful work. We have carefully studied the paper and agree that its "Kimad+" component shares a similar optimization objective with our work: minimizing quantization error under a fixed communication budget. However, our approaches diverge in two fundamental aspects:
>
> * **Distortion Metric:** Kimad uses squared L2-norm of quantization error (a signal-level measure), whereas our LAD metric directly captures the impact of quantization on task loss. Our experiments in **Appendix B.1** consistently show that LAD offers substantially better correlation with downstream performance.
> * **Optimization Algorithm:** Kimad formulates bit allocation as a knapsack problem and solves it using dynamic programming, similar to that of L-GreCo [1]. In contrast, our framework, enabled by the linear superposition property, utilizes a Lagrangian relaxation approach, which finds the optimal solution with linear complexity.
>
> **Reimplementation and Comparison:** As the official source code for Kimad is not available, we have made a best-effort re-implementation of their method based on the description of the paper, specifically using the squared L2-norm distortion metric and a dynamic programming solver. We evaluated it on the ResNet-18/CIFAR-10 benchmark under a 2-bit budget. The results are as follows:
>
> | Method                 | Distortion Metric                    | Top-1 Accuracy (%) |
> | :--------------------- | :----------------------------------- | :-----------------: |
> | Uniform                | -                                    | 77.33              |
> | Greedy                 | MSE (normalized L2-norm squared)     | 87.13              |
> | Kimad (re-implemented) | L2-norm squared                      | 87.22              |
> | Greedy                 | LAD (Ours)                           | 88.09              |
> | **Ours**               | **LAD (Ours)**                       | **88.39**          |
> | **Ours**               | **MSE (normalized L2-norm squared)** | **87.24**          |
>
> This comparison demonstrates a two-fold advantage of our framework. The vast majority of the performance improvement comes from our novel LAD metric. On top of that, our Lagrangian-based optimal solver further refines the bit allocation to achieve the state-of-the-art result.
>
> **References:**
>
> [1] Markov I, Alimohammadi K, Frantar E, et al. L-GreCo: Layerwise-adaptive Gradient Compression For Efficient Data-parallel Deep Learning[J]. Proceedings of Machine Learning and Systems, 2024, 6: 312-324.

---

> ### Author Response · Authors · 2025-11-26
> **Official Response to Reviewer xQTq (Part 3/3)**
>
> **Q2 & Q3: On the Trigger Frequency and the Runtime Cost of the Trigger Mechanism.**
>
> **A5:**
>
> **Trigger Frequency (Q2):** Our proposed trigger is designed to be event-driven and sparse, activating a reallocation only when necessary. For instance, in ResNet-18 on CIFAR-10, the trigger condition was met and a reallocation was performed only 91 times throughout the entire training process of 200 epochs **(Appendix B.2, Table 4)** .
>
> **Runtime Cost of the Trigger Mechanism Itself (Q3):** The trigger is intentionally lightweight, consisting of:
>
> * Per-Layer Norm Calculation: This operation is a single, highly optimized kernel call per layer on a GPU.
> * Vector Construction: Assembling the L2-norm values into a vector is a trivial memory operation.
> * Cosine Similarity Calculation: This is an $\mathcal{O}(L)$ operation.
>
> To provide a concrete figure, we have profiled the  execution time of the trigger in our PyTorch implementation. A single invocation of the trigger mechanism takes **approximately 1.6-1.8 milliseconds** on our experimental hardware (NVIDIA RTX 4090). This confirms that the cost of monitoring the gradient distribution at every step is minimal.
>
>
> ---
>
> **Q4: What are the actual bit allocations obtain by your method and how much do they they differ from Uniform and Greedy?**
>
> **A6:** We have visualized the per-layer bit allocations at different training stages and performed a statistical analysis of their properties in **Appendix D**.
>
> From **Figure 4 in Appendix D**, we can find that our method frequently utilizes intermediate bit-widths (e.g., 3-6 bits). This contrasts with the Greedy approach, which often makes more extreme, "all-or-nothing" decisions—assigning the maximum 8 bits until the budget becomes tight, at which point it can be forced to assign minimal bits to subsequent layers. In addition, **Table 5 in Appendix D** shows that the allocation strategy produced by our method is significantly more stable over the course of training.
>
> *Table 5. Statistical analysis of bit allocation strategies for ResNet-18. Lower values for balance and stability metrics indicate a more balanced and stable allocation strategy.*
>
> | Method                | Balance  (Mean of Std. Dev.) | Stability  (Std. Dev. of Std. Dev.) |
> | :-------------------- | :------------------------------: | :-------------------------------------: |
> | Greedy                |               2.37               |                  0.14                   |
> | **Ours (Lagrangian)** |             **2.26**             |                **0.12**                 |
>
> ---
>
> **Q5: Would your method be able to recover a PPL of around 82? If so, in what configuration?**
>
> **A7:**
>
> Language models are indeed more sensitive to aggressive gradient quantization than vision models due to:
>
> 1. Strong temporal coupling across sequence steps
> 2. Quantization noise contaminating optimizer states (e.g., AdamW momentum/variance)
>
> **Performance Recovery:** Yes, our method can recover the PPL to a level comparable to the FP32 baseline of around 82. Based on our further experiments exploring this trade-off, **our method requires an average bit-width of 5 bits or higher to recover the PPL of around 82**. Our method achieves a PPL of approximately 89 at 4-bit, which is a significant improvement over the 3-bit result and demonstrates a clear and graceful trade-off between compression and performance.
>
> **The configuration** for our dynamic trigger used to obtain these results was: a minimum reallocation interval ($k_{min}$) of 50 iterations, a similarity threshold ($\tau$) of 0.98, and the LAD metric was computed using an expectation over $K=50$ batches.

---

### Author Response · Authors · 2025-11-26
**Review and Rebuttal Summary**

We thank all reviewers for their thorough and constructive feedback. We appreciate the recognition of our work’s key contributions, including (1) introducing a loss-aware distortion (LAD) that aligns quantization with training objectives, (2) formulating gradient quantization as a principled RDO problem with a tractable layer-wise decomposition, and (3) designing an efficient Lagrangian bit-allocation scheme with a dynamic trigger.

We summarize the main concerns and corresponding actions taken during the rebuttal:

Across reviewers, concerns mainly centered on (1) the theoretical assumptions behind LAD and linear superposition, (2) the effect of layer-wise bit reallocation on optimization dynamics and convergence, (3) computational overhead of LAD and the trigger mechanism, and (4) completeness of experimental reporting.

**1. Clarifying Assumptions and Justifying Linear Superposition**

* We added dedicated assumptions paragraphs in **Sec. 2.2.1** and **Appendix A** of revised manuscript, explicitly specifying the smoothness, small-perturbation, and structural orthogonality conditions under which the first-order Taylor approximation and linear superposition hold.

**2. Convergence Analysis and Optimization Dynamics**

- We added a detailed convergence result in **Appendix C.1**, providing finite-step bounds for SGD with LAD-based quantization and discussing how LAD-guided bit reallocation preserves convergence behavior comparable to existing gradient compression methods.
- We established a theoretical link between our objective and the optimization process **(Appendix C.2)**.
- We provided a rigorous derivation proving that the Linear Superposition Property remains valid even when accounting for accumulated quantization errors from previous iterations **(Appendix C.3)**.

**3. Computational Overhead and Practical Efficiency**

* We reported the empirical frequency of regime-shift events and show that overhead is amortized in practice. **(Appendix B.2)**
* We computed detailed **runtime of the proposed trigger (about 1.6-1.8 ms)**.

**4. Additional Results and Completeness of Experiments**

* We added visualizations and statistical analysis of the actual bit-allocation patterns discovered by our method and baselines in **Appendix D**, enabling direct comparison with uniform and greedy strategies.
* We re-implemented and evaluated the "Kimad" method (Xin et al.). Results show our method significantly outperforms it (e.g., **88.39% vs. 87.22%** on ResNet-18/CIFAR-10) due to our superior LAD metric and optimal Lagrangian solver.

We thank the reviewers again for their valuable comments. The revisions in **Sec. 2.2.1**, **Appendix A**, **Appendix C**, and **Appendix D** substantially strengthen the theoretical clarity, empirical transparency, and practical applicability of the work. We hope the updated manuscript addresses all concerns satisfactorily.

---

### Meta-Review · Area_Chair_qvtN · 2026-01-02

**Summary:**

As summarized by authors, across reviewers, concerns mainly centered on (1) the theoretical assumptions behind LAD and linear superposition, (2) the effect of layer-wise bit reallocation on optimization dynamics and convergence, (3) computational overhead of LAD and the trigger mechanism, and (4) completeness of experimental reporting.

Many concerns are answered during the rebuttal. However, in the rebuttal to xQTq, the author did not respond to the query on the computational overhead of computing the loss-aware distortion metric and running the search across all layers.  g5RR and i23A also has similar concerns on computation overhead. During the rebuttal, no explicit comparison regarding computation cost and resource cost is reported. It is hard to understand if the proposed method is cost-effective, which is key for gradient compression. It is suggested to compare the overall training time and memory usage in Table 1 and Table 2 (like Kimad, L-GRECO, AC-SGD, [3], etc.), instead of only reporting model accuracy and compression ratio. Otherwise it is hard to compare different compression methods as there is tradeoff between overhead and performance.

Related work coverage could be improved. Literatures discussed is relatively old, one is published in 2024, none is published in 2023. I can find some closely related works or survey [1][2][3].

Besides, reviewers provide many comments on improving the content organization, theory derivation, and clarity issues, which may need aditional review.

[1] Jia, Jinda, et al. "Sdp4bit: Toward 4-bit communication quantization in sharded data parallelism for LLM training." Advances in Neural Information Processing Systems 37 (2024): 8734-8759.
[2] Hao, Zhiwei, et al. "Low-Precision Training of Large Language Models: Methods, Challenges, and Opportunities." arXiv preprint arXiv:2505.01043 (2025).
[3] Choi, Dahun, and Hyun Kim. "GradQ-ViT: Robust and Efficient Gradient Quantization for Vision Transformers." Proceedings of the AAAI Conference on Artificial Intelligence. Vol. 39. No. 15. 2025.

**Reviewer Concerns:**

solved:
The proposed method targets weight and gradient compression, which does not address the core memory/compute challenges of training very large models. Its applicability also appears limited to data parallelism.
Comparison with Kimad: Adaptive Gradient Compression with Bandwidth Awareness.
The dynamic reallocation trigger increases deployment complexity.
Lacks a dedicated Related Work section.
The paper contains few experimental figures, and the presentation of results is not sufficiently intuitive.
Needs convergence guarantees and explicit bounds.
Practical training often exhibits nonlinearity and interaction between layers.
Some other detailed concerns on math derivations.




still outstanding concerns:
Do the end-to-end speedups from compression outweigh the method’s computational overhead? Is it more efficient than a strong greedy compression baseline that achieves similar quality? (xQTq).
Evaluate larger models such as ViT-Large and BERT-Large (xQTq).
The Loss-Aware Distortion metric's resource cost for large-scale models remains unmeasured, risking computational bottlenecks. (g5RR).
Is the improvement worth the additional computational cost? (i23A).

**Reviewer Scores:**

xQTq might keep 2 as xQTq's concern on cost-effectiveness and evaluating on larger models are not solved.
g5RR might keep 8.
i23A might keep 4 or increase as part of his concerns are solved. i23A is also concerned on the cost-effectiveness, while the computational cost is not clarified during rebuttal.
GMhX might keep 4 or increase as part of his concerns are solved.

---

### Decision · Program_Chairs · 2026-01-26

Reject